# The *Listeria monocytogenes* persistence factor ClpL is a potent stand-alone disaggregase

Valentin Bohl[1], Nele Merret Hollmann[2], Tobias Melzer[1], Panagiotis Katikaridis[1], Lena Meins[1], Bernd Simon[2], Dirk Flemming[3], Irmgard Sinning[3], Janosch Hennig[2,4], Axel Mogk[1]*

[1]Center for Molecular Biology of Heidelberg University (ZMBH), DKFZ-ZMBH Alliance, Heidelberg, Germany; [2]Structural and Computational Biology Unit, European Molecular Biology Laboratory (EMBL) Heidelberg, Heidelberg, Germany; [3]Heidelberg University Biochemistry Center (BZH), Heidelberg, Germany; [4]Chair of Biochemistry IV, Biophysical Chemistry, University of Bayreuth, Bayreuth, Germany

**\*For correspondence:**
a.mogk@zmbh.uni-heidelberg.de

**Competing interest:** The authors declare that no competing interests exist.

**Abstract** Heat stress can cause cell death by triggering the aggregation of essential proteins. In bacteria, aggregated proteins are rescued by the canonical Hsp70/AAA+ (ClpB) bi-chaperone disaggregase. Man-made, severe stress conditions applied during, e.g., food processing represent a novel threat for bacteria by exceeding the capacity of the Hsp70/ClpB system. Here, we report on the potent autonomous AAA+ disaggregase ClpL from *Listeria monocytogenes* that provides enhanced heat resistance to the food-borne pathogen enabling persistence in adverse environments. ClpL shows increased thermal stability and enhanced disaggregation power compared to Hsp70/ClpB, enabling it to withstand severe heat stress and to solubilize tight aggregates. ClpL binds to protein aggregates via aromatic residues present in its N-terminal domain (NTD) that adopts a partially folded and dynamic conformation. Target specificity is achieved by simultaneous interactions of multiple NTDs with the aggregate surface. ClpL shows remarkable structural plasticity by forming diverse higher assembly states through interacting ClpL rings. NTDs become largely sequestered upon ClpL ring interactions. Stabilizing ring assemblies by engineered disulfide bonds strongly reduces disaggregation activity, suggesting that they represent storage states.

## eLife assessment

This **important** manuscript details the characterization of ClpL from *L. monocytogenes* as an effective and autonomous AAA+ disaggregase that provides enhanced heat resistance to this food-borne pathogen. Supported by **compelling** evidence, the authors demonstrate that ClpL has DnaK-independent disaggregase activity towards a variety of aggregated model substrates, which is more potent than that observed with the endogenous canonical DnaK/ClpB bi-chaperone system. The work will be of broad interest to microbiologists and biochemists.

## Introduction

*Listeria monocytogenes* (*Lm*) is a food-borne pathogen and the causative agent of listeriosis, a severe human illness that is acquired upon eating contaminated food with a mortality of 20–30% (*Bucur et al., 2018*). While *Lm* can efficiently persist in food-producing plants, it cannot grow at temperatures above 45°C, qualifying heat treatments as efficient possibility to eradicate the pathogen. However, *Lm* strains that show an up to 1000-fold enhanced heat resistance have been described (*Lundén*

*et al., 2008*). Environmental and food-related *Lm* strains frequently harbor plasmids that encode for open reading frames providing protection against antibiotics, disinfectants, heavy metals, high salt concentrations, low pH, and heat stress (*Jiang et al., 2016*; *Lebrun et al., 1994*; *Naditz et al., 2019*; *Pöntinen et al., 2017*; *Poyart-Salmeron et al., 1990*). Heat resistance at 55°C is provided by the chaperone ClpL, which is encoded on diverse *Lm* plasmids (*Pöntinen et al., 2017*). ClpL presence does not increase the maximum growth temperature of *Lm* but strongly enhances cell survival upon sudden exposure to extreme temperatures. ClpL is also encoded in the core genome of various Gram-positive bacteria including the major pathogens *Staphylococcus aureus* and *Streptococcus pneumoniae*. *clpL* gene expression is typically upregulated upon heat shock and *clpL* knockout strains can exhibit increased sensitivity to heat stress, suggesting a function of ClpL in cellular proteostasis (*Li et al., 2011*; *Suokko et al., 2008*; *Suokko et al., 2005*).

ClpL belongs to the AAA+ (<u>A</u>TPase <u>a</u>ssociated with diverse cellular <u>a</u>ctivities) protein family. AAA+ proteins typically form hexameric assemblies and thread substrates through an inner channel under consumption of ATP. This threading activity of AAA+ proteins is linked to unfolding and disassembly of substrates (*Puchades et al., 2020*). In vitro ClpL, originating from other Gram-positive bacteria, was shown to exhibit chaperone activity, however, which specific role ClpL plays in bacterial proteostasis remained undefined (*Kwon et al., 2003*; *Park et al., 2015*; *Tao and Biswas, 2013*).

Severe heat shock triggers massive protein aggregation, titrating crucial components of essential cellular processes including transcription, translation, and cell division. Cells are therefore equipped with protein disaggregases that rescue aggregated proteins and protect cells from the deleterious consequences of protein aggregation. In bacteria the AAA+ protein ClpB cooperates with the Hsp70 (DnaK) chaperone to form the canonical and widespread bi-chaperone disaggregation system (*Mogk et al., 1999*; *Zolkiewski, 1999*). Notably, ClpB does not directly bind to protein aggregates but requires Hsp70 for aggregate targeting (*Winkler et al., 2012*). The DnaK/ClpB disaggregation activity is well adapted to temperature gradients, which allow for increasing disaggregation capacity through induction of stress responses. Surprisingly, ClpB is equipped with a reduced unfolding power (*Haslberger et al., 2008*), limiting its disaggregation efficiency in vitro toward tight aggregates forming at high unfolding temperatures (*Katikaridis et al., 2019*).

Bacteria face new stress conditions in the industrial world including abrupt temperature jumps applied during, e.g., food processing to reduce the number of bacterial contaminants. Selected Gram-negative bacteria including major pathogens like *Pseudomonas aeruginosa* (*Pa*) or *Klebsiella pneumoniae* adapt to these man-made stress regimes by acquiring the autonomous AAA+ ClpG disaggregase, which confers extreme heat resistance (*Bojer et al., 2010*; *Boll et al., 2017*; *Lee et al., 2018*). *clpG* is encoded on plasmids or mobile genomic islands (transmissible locus of stress resistance [tLST]) together with other open reading frames linked to protein quality control (*Kamal et al., 2021b*). The tLST cluster can be transferred to other bacteria via horizontal gene transfer. ClpG homologs are not present in Gram-positive bacteria and a disaggregase counterpart sharing mechanistic features has not been described yet. The characteristics of *Lm* ClpL, including its link to superior heat resistance and its presence on conjugative plasmids (*Pöntinen et al., 2017*), show remarkable similarity to ClpG.

Here, we characterize *Lm* ClpL as autonomous and general, robust disaggregase. This defines ClpL as counterpart of ClpG in Gram-positive bacteria. ClpL has higher disaggregation activity as compared to the canonical *Lm* DnaK/ClpB disaggregase by applying enhanced threading forces. Furthermore, ClpL shows increased thermostability compared to *Lm* DnaK, enabling it to withstand more severe heat stress conditions. ClpL achieves aggregate specificity via a unique N-terminal domain (NTD), which features mobile secondary structure elements and interacts with protein aggregates via patches of aromatic residues. The comparison of ClpL and ClpG allows to define mechanistic features shared by stand-alone bacterial disaggregases.

## Results

### *Lm* ClpL is an autonomous disaggregase

*Lm* ClpL increases heat resistance and is encoded on conjugative plasmids, thus sharing characteristic features with the autonomous, potent disaggregase ClpG from selected Gram-negative bacteria. ClpL also has a similar domain organization as compared to the stand-alone *Pa* ClpG and

the Hsp70-dependent ClpB disaggregase, including two AAA domains, a coiled-coil M-domain (MD), and the absence of a ClpP interaction motif, implying a function independent of protein degradation (*Figure 1A*, *Figure 1—figure supplement 1*). We therefore asked whether ClpL represents a stand-alone disaggregase, functioning as counterpart of ClpG in Gram-positive bacteria. ClpL showed substantial disaggregation activity toward heat-aggregated luciferase that was 0.67-fold (p=1.9e-16) reduced as compared to ClpG, but 3.1-fold higher as compared to the *Lm* Hsp70 (KJE)-ClpB bi-chaperone disaggregase (p=1e-23), which we included as references (*Figure 1B*). Disaggregation activity of ClpL was dependent on ATP and not observed in the presence of ADP or absence of nucleotide (*Figure 1—figure supplement 2A*). Addition of *Lm* KJE to ClpL increased the reactivation of aggregated luciferase 1.7-fold (p=1.7e-18), however, it did not significantly enhance ClpL disaggregation activity ($p_{1 \times KJE}$ = 0.19, $p_{2 \times KJE}$ = 0.19) when monitored directly by turbidity measurements (*Figure 1—figure supplement 2B/C*). This argues against a direct cooperation of *Lm* ClpL and KJE in disaggregation and points to a role of *Lm* KJE in supporting refolding of disaggregated luciferase.

We next tested whether ClpL acts as a general, robust disaggregase by monitoring disaggregation of other heat-aggregated model substrates: GFP, α-glucosidase, and malate dehydrogenase (MDH). ClpL always showed high disaggregation activity that was similar to *Pa* ClpG and vastly superior to *Lm* KJE/ClpB (*Figure 1C*, *Figure 1—figure supplement 2D/E*). Combining ClpL with *Lm* KJE led to a strong reduction in GFP, α-glucosidase, and MDH disaggregation activity, which can be explained by competition for protein aggregate binding, as observed for ClpG before (*Lee et al., 2018*). We also compared *Lm* ClpL disaggregation activity with the *E. coli* (*Ec*) KJE/ClpB system, documenting similar (luciferase, MDH) or approx. 2.5-fold enhanced (GFP [p=2.8e-5], α-glucosidase [p=6.1e-6]) activities (*Figure 1—figure supplement 2F*). Addition of *Ec* KJE to ClpL reduced or abrogated protein disaggregation (*Figure 1—figure supplement 2F*). The presence of the *Lm* or *Ec* DnaK systems reduced binding of ClpL to MDH aggregates (*Figure 1—figure supplement 2G*), indicating that the inhibitory effects stem from competition for aggregate binding. Notably, luciferase aggregates differed from all other aggregated model substrates tested, as a strong inhibition of ClpL disaggregation activity was not observed in the presence of the DnaK systems. This points to specific structural features of luciferase aggregates or the presence of distinct binding sites on the aggregate surface that favor ClpL binding.

We next asked whether ClpL from other Gram-positive bacteria also exhibits high disaggregation activity. We selected *L. gasseri* (*Lg*) ClpL, as this bacterium does not encode for the canonical disaggregase ClpB and Δ*clpL* cells exhibit a loss of heat resistance (*Figure 1D*; *Suokko et al., 2008*). *Lg* ClpL exhibited high disaggregation activity toward aggregated luciferase and MDH that was similar to *Lm* ClpL (*Figure 1E*). This explains why *Lg* Δ*clpL* cells are heat-sensitive, as they lost the central disaggregase.

Together our findings establish ClpL as potent, autonomous disaggregase, whose disaggregation activity is comparable to ClpG. Our findings are different from former, initial analyses on ClpL from *Streptococcus* sp. reporting on either no (*Tao and Biswas, 2013*) or low disaggregation activity (*Park et al., 2015*) toward single model substrates tested.

## Specific features distinguish ClpL from the KJE/ClpB disaggregase

*Lm* ClpL exhibits superior disaggregation activity as compared to *Lm* KJE/ClpB. We speculated that ClpL applies a higher threading power, enabling it to process tight protein aggregates, which are largely resistant to KJE/ClpB activity. We made use of the fusion construct luciferase-YFP, which forms mixed aggregates at 46°C that consist of unfolded luciferase and native YFP moieties (*Figure 2A*). Threading power can be assessed by monitoring YFP fluorescence during the disaggregation process. *Ec* KJE/ClpB and *Lm* KJE/ClpB did not unfold YFP during the disaggregation process, documenting limited unfolding power, consistent with former reports (*Haslberger et al., 2008*; *Katikaridis et al., 2019*). In contrast, we observed a rapid loss of YFP fluorescence in the presence of ClpL that was even faster as compared to ClpG (*Figure 2B*). We conclude that stand-alone disaggregases ClpL and ClpG but not the canonical KJE/ClpB disaggregase exhibit robust threading activities that allow for unfolding of tightly folded domains.

ClpL strongly enhances survival of *Lm* cells at 55°C, a temperature that is far beyond the maximum growth temperature (45°C) of *Lm* cells. We speculated that these high temperatures inactivate KJE/ClpB activity by thermal unfolding of one of the chaperones involved. We therefore determined the

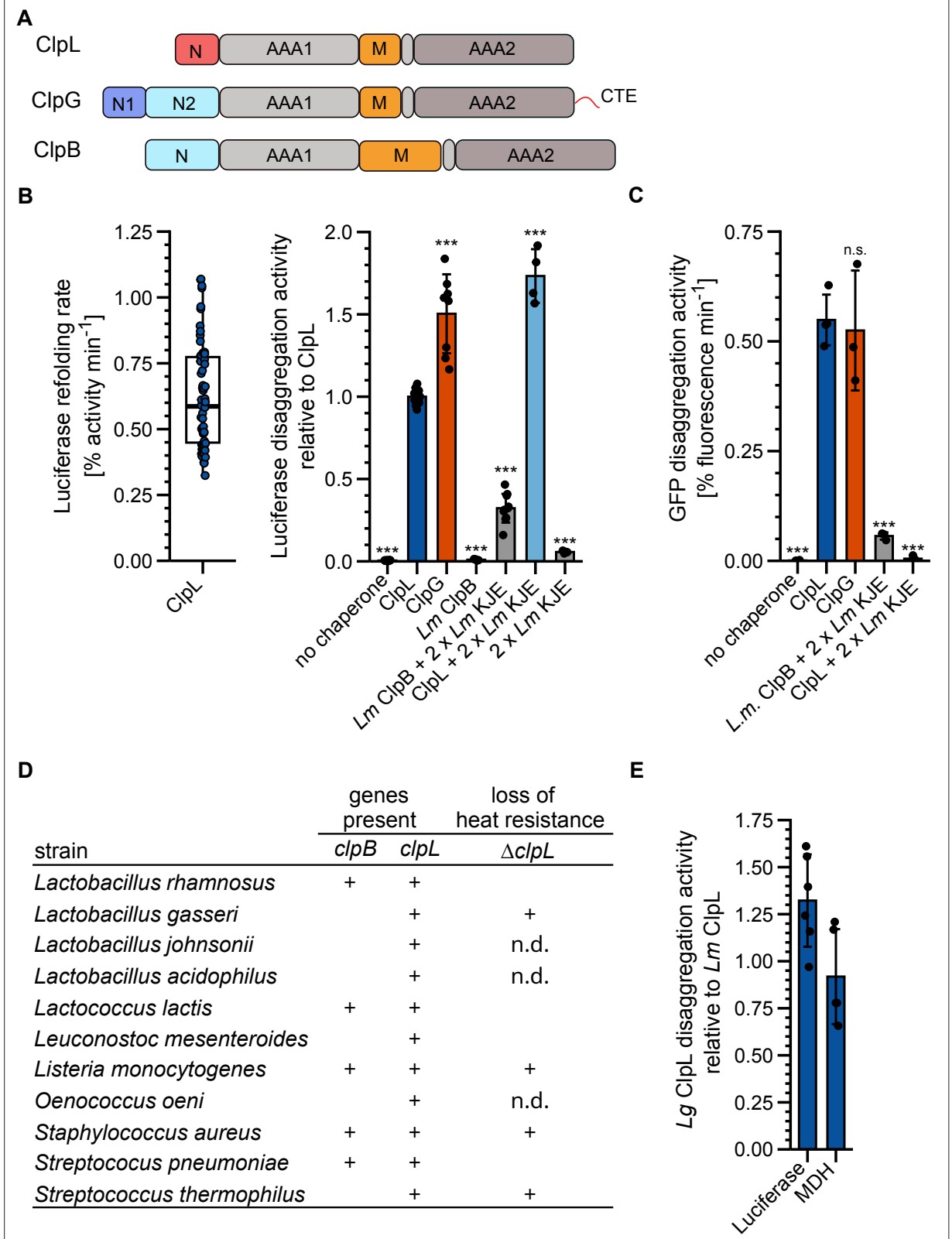

**Figure 1.** ClpL is an autonomous disaggregase. (**A**) Domain organizations of ClpL, ClpG, ClpB. All AAA+ proteins consist of two AAA domains (AAA1, AAA2), a coiled-coil middle domain (**M**) and diverse N-terminal domains (**N**). ClpG additionally harbors a disordered C-terminal extension (CTE). (**B**) Left: Luciferase disaggregation activity (% refolding of aggregated luciferase/min) of *Lm* ClpL was determined. Right: Relative luciferase disaggregation activities of indicated disaggregation systems were determined. KJE: DnaK/DnaJ/GrpE. The disaggregation activity of *Lm* ClpL was set to 1. (**C**) GFP

*Figure 1 continued on next page*

*Figure 1 continued*

disaggregation activities (% refolding of aggregated GFP/min) of indicated disaggregation systems were determined. (**D**) Occurrence of ClpB and ClpL disaggregases in selected Gram-positive bacteria. Loss of heat resistance upon *clpL* deletion in bacteria harboring solely ClpL is indicated. n.d.: not determined. (**E**) Relative luciferase and malate dehydrogenase (MDH) disaggregation activities (each: % regain of enzymatic activity/min) of *Lactobacillus gasseri* (*Lg*) ClpL. Disaggregation activities of *Lm* ClpL were set to 1. Shown is a boxplot (**B**) or data points and mean ± SD (B/C/E), n≥3. Statistical analysis: one-way ANOVA, Welch's test for post hoc multiple comparisons. Significance levels: *p<0.05; **p<0.01; ***p<0.001. n.s.: not significant.

The online version of this article includes the following figure supplement(s) for figure 1:

**Figure supplement 1.** Sequence alignment of *Escherichia coli* ClpB, *S. aureus* ClpC, *P. aeruginosa* ClpG$_{GI}$, and *L. monocytogenes* ClpL.

**Figure supplement 2.** ClpL is a potent, stand-alone disaggregase.

thermal stabilities of ClpL and the central components DnaK and ClpB of the bi-chaperone disaggregase by monitoring protein unfolding during temperature increase via SYPRO Orange binding and nanoDSF (*Figure 2C/D*). *Lm* ClpL and ClpB exhibited comparable melting temperatures of 60–63°C, while the thermal stability of *Lm* DnaK was much lower (T$_M$: 47–50°C) (*Figure 2C/D*). Notably, unfolding of DnaK at 55–58°C was reversible, as preincubation of the chaperone at the heat shock temperatures did not abolish DnaK-dependent disaggregation activity at 30°C (*Figure 2—figure supplement 1*). We infer that incubation at 55°C will cause unfolding of *Lm* DnaK, thereby abrogating KJE/ClpB disaggregation activity. The higher thermal stability of ClpL will provide disaggregation power to *Lm* cells at 55°C, explaining its strong protective effect on bacterial survival.

## The NTD of ClpL mediates aggregate targeting

We explored the mechanistic basis of ClpL disaggregation activity. We first tested for roles of its AAA1/2 domains by mutating glutamate residues of the Walker B motifs (E197A, E520A), crucial for ATP hydrolysis, and aromatic pore loop residues (Y170A, Y504A), involved in substrate threading (*Figure 3—figure supplement 1A*). ClpL wild-type (WT) exhibits high, concentration-independent ATPase activity (90.1±26.1 ATP/min/monomer) (*Figure 3—figure supplement 1B*). ATPase activities of pore loop mutants were similar to WT, while they were strongly reduced in Walker B mutants (*Figure 3—figure supplement 1C*). A ClpL pore-1 loop mutant (Y170A) retained partial disaggregation activity, while blocking ATP hydrolysis in one AAA domain (E197A, E520A) or mutating the pore-2 site (Y504A) abrogated disaggregation (*Figure 3—figure supplement 1D*). These findings are similar to phenotypes of corresponding ClpB and ClpG mutants and underline that two functional AAA domains and an intact pore-2 site are crucial for disaggregation activity.

We next turned our interest to the NTD of ClpL, which is distinct from the NTDs of ClpG and ClpB (*Figure 1—figure supplement 1*). NTDs are connected via flexible linkers to the AAA ring structure and typically mediate substrate or partner specificity (*Kirstein et al., 2009*). We deleted the NTD (ΔN: ΔA2-N61) of ClpL and observed a total loss of disaggregation activity toward luciferase and MDH aggregates (*Figure 3A*). The ATPase activity of ΔN-ClpL was reduced (65% of ClpL-WT, p=5.8e-4) (*Figure 3B*). However, this loss was disproportionally much smaller (*Figure 3B*), arguing against defects in ATP hydrolysis caused by, e.g., basic structural defects as reason for disaggregation activity loss. To explore the role of the NTD for disaggregation activity in an in vivo setting, we expressed ClpL and ΔN-ClpL in *Ec ΔclpB* cells and monitored the recovery of aggregated luciferase (*Figure 3C*). Expression of *clpL* but not *ΔN-clpL* allowed for regain of luciferase activity (16.5% [WT] vs 1.6% [ΔN-ClpL] after 120 min at 30°C). ClpL disaggregation activity was lower than *Pa* ClpG activity but higher than *Ec* ClpB activity, which served as references. Expression of *clpL*, but not *ΔN-clpL*, also restored thermotolerance in *Ec ΔclpB* and *dnaK103* mutant cells, lacking a functional DnaK chaperone (*Figure 3D/E*, *Figure 3—figure supplement 2A*). Expression levels of disaggregases in *ΔclpB* and *dnaK103* were comparable (*Figure 3—figure supplement 2B/C*). These data underline the robust, autonomous disaggregation activity of ClpL and document the essential role of its NTD.

This role of the NTD in aggregate targeting can explain the defect of ΔN-ClpL in protein disaggregation. To directly demonstrate this function, we fused the ClpL NTD to ΔN-ClpB-K476C (ΔN-ClpB*) generating L$_N$-ClpB* (*Figure 3F*). The ClpB NTD mediates binding to soluble unfolded proteins (*Rosenzweig et al., 2015*), however, it is dispensable for disaggregation activity (*Iljina et al., 2021*; *Mogk et al., 2003*). The K476C mutation is located in the ClpB MD, abrogating ClpB repression and allowing for high ATPase activity in the absence of DnaK. L$_N$-ClpB* exhibited high disaggregation activity, whereas ClpB* and ΔN-ClpB* relied on the presence of the cooperating DnaK (KJE) system

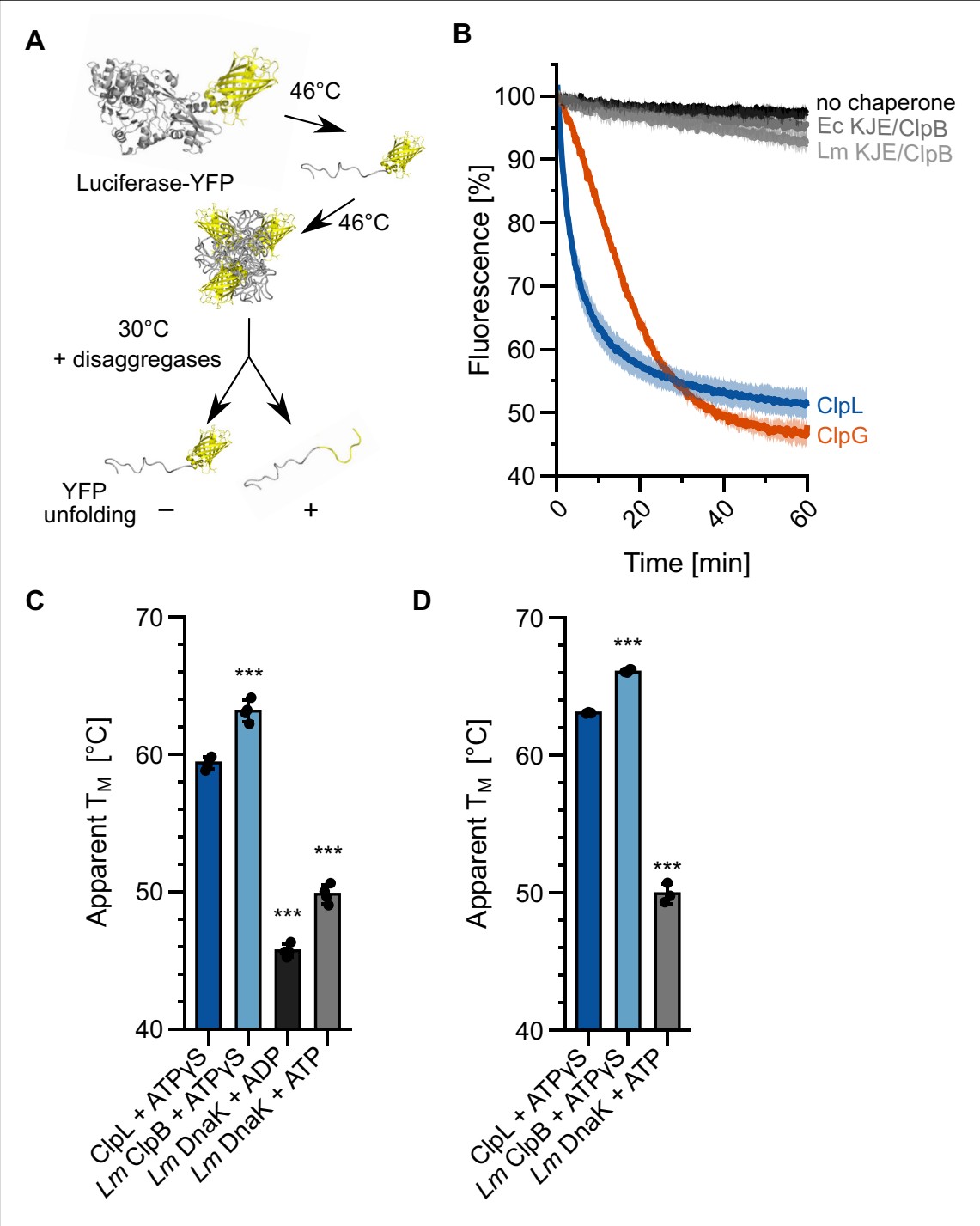

**Figure 2.** Specific molecular features separate ClpL from ClpB/DnaK. (**A**) Incubation of luciferase-YFP at 46°C only leads to unfolding of the luciferase moiety and the formation of mixed aggregates including folded YFP. Unfolding of YFP during disaggregation of aggregated luciferase-YFP reports on threading power. (**B**) Aggregated luciferase-YFP was incubated in the presence of indicated disaggregation machineries and YFP fluorescence was recorded. Initial YFP fluorescence was set at 100%. (C/D) Melting temperatures of ClpL, *L. monocytogenes* (*Lm*) ClpB, and *Lm* DnaK were determined in the presence of indicated nucleotides by SYPRO Orange binding (**C**) or nanoDSF (**D**). Shown are mean curves ± SD (**B**) or data points and mean ± SD (C/D), n≥3. Statistical analysis: one-way ANOVA, Welch's test for post hoc multiple comparisons. Significance levels: *p<0.05; **p<0.01; ***p<0.001. n.s.: not significant.

The online version of this article includes the following figure supplement(s) for figure 2:

**Figure supplement 1.** Unfolding of DnaK at high temperatures is reversible.

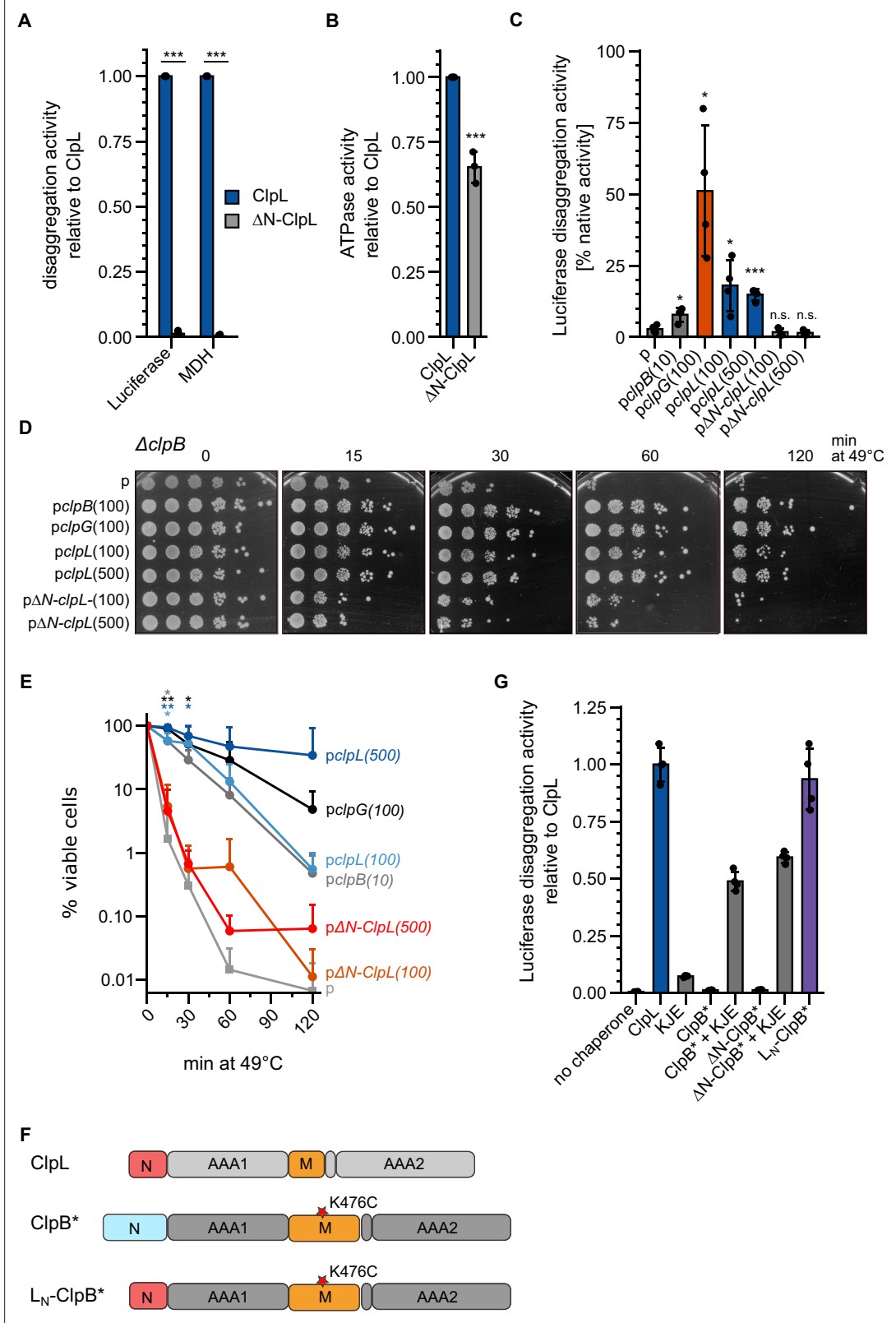

**Figure 3.** The ClpL N-terminal domain (NTD) is essential for disaggregation activity. (**A**) Disaggregation activities of ClpL and ΔN-ClpL toward aggregated luciferase and malate dehydrogenase (MDH) (each: % regain enzymatic activity/min) were determined. The activity of ClpL was set to 1. (**B**) ATPase activities of ClpL and ClpL-ΔN were determined. The ATPase activity of ClpL was set to 1. (**C**) *E. coli* Δ*clpB* cells harboring plasmids for constitutive expression of luciferase and IPTG-controlled expression of indicated disaggregases were grown at 30°C to mid-logarithmic growth phase

*Figure 3 continued on next page*

*Figure 3 continued*

and heat shocked to 46°C for 15 min. Luciferase activities prior to heat shock were set to 100%. The regain of luciferase activity was determined after a 120 min recovery period at 30°C. 10/100/500: μM IPTG added to induce disaggregase expression. p: empty vector control. (**D**) *E. coli ΔclpB* cells harboring plasmids for expression of indicated disaggregases were grown at 30°C to mid-logarithmic growth phase and shifted to 49°C. Serial dilutions of cells were prepared at the indicated time points, spotted on LB plates, and incubated at 30°C. 10/100/500: μM IPTG added to induce disaggregase expression. p: empty vector control. (**E**) Quantification of data from (**D**). (**F**) Domain organization of the ClpL-ClpB chimera L$_N$-ClpB*. ClpB* harbors the K476C mutation in its M-domain, abrogating ATPase repression. (**G**) Relative luciferase disaggregation activities (% refolded luciferase/min) of indicated disaggregation machineries were determined. The disaggregation activity of ClpL was set to 1. Shown are mean curves and data points (**E**) or data points and mean ± SD (A/B/C/G), n≥3. Statistical analysis: one-way ANOVA, Welch's test for post hoc multiple comparisons. Significance levels: *p<0.05; **p<0.01; ***p<0.001. n.s.: not significant.

The online version of this article includes the following source data and figure supplement(s) for figure 3:

**Figure supplement 1.** ClpL disaggregation activity relies on ATP-fueled substrate threading.

**Figure supplement 2.** The ClpL N-terminal domain (NTD) is crucial for disaggregation activity.

**Figure supplement 2—source data 1.** Source data include non-cropped and non-processed images of SDS-gels and western blots.

**Figure supplement 2—source data 2.** Source data include non-cropped and non-processed images of SDS-gels and western blots and indicate sections and loading schemata of the respective figure supplement.

(*Figure 3G*). Thus, fusion of ClpL NTD converts ClpB into a stand-alone disaggregase bypassing the need of DnaK for aggregate targeting. Notably, L$_N$-ClpB* still exerted a reduced threading power as compared to ClpL revealed by strong differences in YFP unfolding activities when using aggregated luciferase-YFP as substrate (*Figure 3—figure supplement 2D*). This suggests that the AAA threading motors and the aggregate-targeting NTD largely function independently.

## Molecular principle of aggregate recognition by ClpL

To understand how the ClpL NTD selectively interacts with protein aggregates, we predicted its structure with AlphaFold2 (*Jumper et al., 2021*; *Mirdita et al., 2022*) and validated this model by nuclear magnetic resonance spectroscopy (NMR). AlphaFold2 predicts a conformation of two α-helices (α1, α2) and a short anti-parallel β-sheet, while N- and C-terminal regions are disordered (*Figure 4A*). The β-sheet forms a small hydrophobic core with α2, which in turn is predicted to interact with α1, forming a second small core structure (*Figure 4A*). The latter hydrophobic core is formed mostly by π-stacking interactions of four aromatic residues (F19, F23, Y36, and F48, *Figure 4B*) and the side chain methyl group of A34. The C-terminal hydrophobic core is formed by residues Y36, V38, L43, F48, Y51, and L57. Thus, Y36 and F48 would take part in the formation of both hydrophobic cores. The $^{15}$N-HSQC NMR spectrum of the NTD shows several well-dispersed resonances, indicative of a folded domain (*Figure 4—figure supplement 1A*). NMR secondary chemical shifts based on Cα and Cβ backbone resonance assignments confirm the formation of α2 and the β-sheet and thus validate partly the structure prediction (*Figure 4B*). However, although a certain helical propensity can be confirmed from secondary chemical shifts for α1, the formation seems rather transient compared to α2. While NOEs between relevant residues confirm the existence of the C-terminal hydrophobic core (*Figure 4—figure supplement 1B/C*), long-range NOEs expected to arise based on the predicted tertiary interactions between α1 and α2 are entirely missing and only intra-residue or sequential NOEs are visible (*Figure 4—figure supplement 1B and D*). This indicates that the ClpL NTD forms one predicted folded core structure but that α1 forms only transiently and does not interact with α2. To further confirm this, we acquired $^{15}$N spin relaxation data to measure residue-wise dynamics in the ps-ns time scale (*Figure 4—figure supplement 1E*). Longitudinal (R$_1$) and transverse (R$_2$) relaxation rates and heteronuclear NOE data clearly show that the N-terminal region up to residue A34 is more flexible than the hydrophobic core formed by α2 and the β-sheet. The overall low heteronuclear NOE values, rarely above 0.6, also indicate that the secondary structure elements (α1 vs. α2/β-sheet) are mobile with respect to each other in the ps-ns time scale. These observations are consistent with pLDDTs that are generally <70 and partially <50 in AlphaFold2 predictions (*Figure 4A*). Thus, the NTD is highly mobile and formed by a hydrophobic core between secondary structure elements α2 (D46-T54) and the β-sheet formed by residues R35-V38 and Q41-T44. The α-helix 1 forms only transiently and is mobile, i.e., does not interact with α2, thereby making several aromatic residues available for substrate binding.

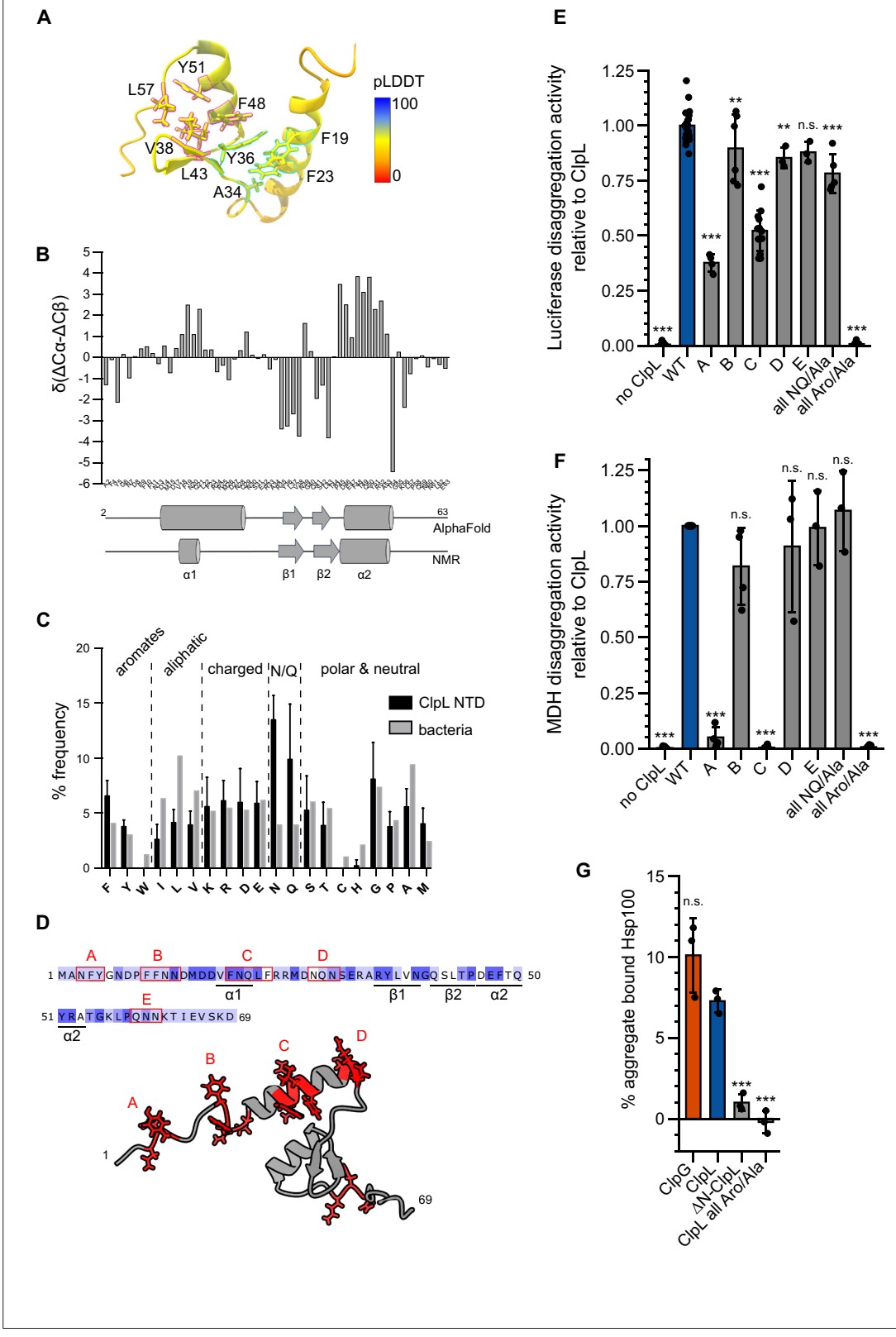

**Figure 4.** Molecular basis of ClpL N-terminal domain (NTD) binding to protein aggregates. (**A**) AlphaFold2 model of ClpL NTD. The color code depicts the calculated confidence of the prediction (pLDDT). Residues that potentially participate in the formation of small hydrophobic cores (α1/2 [cyan] and α2/β1,2 [magenta]) are indicated. (**B**) Secondary structure of ClpL-NTD as determined by magnetic resonance spectroscopy (NMR) using

*Figure 4 continued on next page*

*Figure 4 continued*

secondary chemical shifts (Cα, Cβ). Secondary structure elements determined from NMR and from the AlphaFold prediction are indicated below the histogram. The predicted α1-helix only transiently forms in isolated solution context, which is confirmed by further NMR analysis (*Figure 4—figure supplement 1E*). (**C**) Composition of ClpL NTDs. The frequencies (%) of individual amino acids represent the ratio of the number of a particular residue and the total length of respective NTDs (*L. monocytogenes, S. aureus, S. pneumoniae, Lactobacillus plantarum, Oenococcus oeni, Lactobacillus rhamnosus, Streptococcus suis*). The average frequency of each amino acid in the total bacterial proteomes is given as reference (*Bogatyreva et al., 2006*). (**D**) Localization of patches A-E consisting of aromatic and N/Q residues are indicated. (**E/F**) Luciferase and MDH disaggregation activities (% refolded enzyme/min) of ClpL wild-type (WT) and indicated patch mutants were determined. The disaggregation activity of ClpL was set to 1. (**G**) ClpG, ClpL, and indicated ClpL mutants were incubated with aggregated MDH in the presence of ATPγS. The extend of aggregate binding was determined by co-sedimentation upon centrifugation and quantifications of chaperone levels in soluble and insoluble fractions. Shown are data points and mean ± SD (C/E/F/G), n≥3. Statistical analysis: one-way ANOVA, Welch's test for post hoc multiple comparisons. Significance levels: *p<0.05; **p<0.01; ***p<0.001. n.s.: not significant.

The online version of this article includes the following figure supplement(s) for figure 4:

**Figure supplement 1.** Structural analysis of ClpL N-terminal domain (NTD).

**Figure supplement 2.** Mutant analysis of ClpL N-terminal domain (NTD).

---

Sequence analysis of multiple ClpL NTDs revealed a strongly increased frequency of asparagine (N) and glutamine (Q) residues, while aliphatic residues were less abundant compared to their average occurrence in bacterial proteomes (*Figure 4C*). These sequence features can explain the lack of a stable compact NTD structure. We also noticed an increased frequency of phenylalanine residues. We found aromatic and N/Q residues are frequently located together in patches A-E, which are all located outside the small α2/β-sheet core (*Figure 4D*). To explore the role of those patches and of the high aromatic and N/Q content for substrate binding and protein disaggregation activity, we replaced aromatic and N/Q residues of individual patches with alanines (e.g. patch A: 3NFY5 to 3AAA5). Alternatively, we mutated all aromatic or all N/Q residues present in patches A-E to alanines (all Aro/Ala, all NQ/Ala). All ClpL NTD mutants showed high ATPase activities, excluding severe assembly defects or direct effects on ATP hydrolysis (*Figure 4—figure supplement 2A*). We only observed a 3.6-fold reduced ATPase activity (p=6.7e-12) for the 'all Aro/Ala' construct at low protein concentrations (0.125 µM). At the concentration used in the disaggregation assay (1 µM), no differences to ClpL-WT were determined (p=1) (*Figure 4—figure supplement 2A*).

All NTD mutants were tested for disaggregation activity toward aggregated luciferase and MDH (*Figure 4E/F*). Replacement of aromatic residues (all Aro/Ala) abrogated disaggregation while mutating all N/Q residues had no effect (*Figure 4—figure supplement 2A*). Patch A and C mutants exhibited reduced (luciferase, $p_A$ = 3.6e-34; $p_C$ = 2.6e-41) or lost (MDH, $p_A$ = 1.7e-17; $p_C$ = 6.9e-21) disaggregation activities. The diverse extents of disaggregation activity loss toward luciferase and MDH aggregates point to differences in aggregate structures or distributions of ClpL NTD binding sites. Furthermore, the mutated NTD residues might make specific contributions to aggregate recognition, which differ for the aggregated model substrates tested.

The loss of disaggregation activity of selected NTD mutants (ΔN, all Aro/Ala) could be linked to a deficiency in binding aggregated luciferase (*Figure 4G*), indicating a crucial role of aromatic residues in aggregate recognition. In an additional set of experiments, we combined alternative patch A-C mutants, in which only the aromatic residues were mutated and generated single point mutations of aromatic residues being part of the α2/β-sheet core (Y36A, F48A, Y51A; *Figure 4—figure supplement 2B*). All patch combinations showed aggravated disaggregation activities as compared to single patch mutants. Furthermore, disaggregation activity of ClpL-Y51A was severely affected (for luciferase: 20 ± 12% of WT activity [$p_{Luc}$ = 1.6e-33], for MDH: 5 ± 5% of WT [$p_{MDH}$ = 2.6e-8]), but not for Y36A (for luciferase: 53 ± 8% of WT activity [$p_{Luc}$ = 2.9e-19], for MDH: 68 ± 32% of WT [$p_{MDH}$ = 8.3e-4]) and F48 (for luciferase: 48 ± 6% of WT activity [$p_{Luc}$ = 1.4e-21], for MDH: 70 ± 20% of WT [$p_{MDH}$ = 1.1e-3]) (*Figure 4—figure supplement 2B/C*). None of these mutants were affected in ATP hydrolysis (*Figure 4—figure supplement 2D*). Together these data hint at specific contributions of selected NTD residues in aggregate recognition. ClpL NTD mutants might have additional effects on disaggregation activity by, e.g., controlling substrate transfer to the processing pore site.

We finally recapitulated the effect of key NTD mutations in the $L_N$-ClpB* fusion construct. We confirmed the crucial function of aromatic residues in patches A-C, yet the Y51A ($p_{Luc}$ = 2.3e-12; $p_{MDH}$ = 0.63) did not exhibit the same severe phenotype as observed for ClpL, suggesting a context-specific role (*Figure 4—figure supplement 2E/F*). Defects in protein disaggregation were again linked to defects in aggregate binding, whereas ATPase activities of all mutants tested were similar to the $L_N$-ClpB* reference (*Figure 4—figure supplement 2G/H*). Together these findings indicate an important role of aromatic residues in the flexible N-terminal region of ClpL for disaggregation activity. Our data suggest that the total number of aromatic residues but also their particular identity represent crucial parameters for substrate binding.

We next determined how many NTDs must be present in a ClpL ring to achieve high disaggregation activity. We speculated that ClpL might utilize multiple NTDs to simultaneously contact various binding sites present in close vicinity on an aggregate surface. Such recognition principle would specifically target ClpL to an aggregate while preventing ClpL action on soluble, non-native polypeptides (e.g. nascent polypeptide chains). We performed mixing experiments using $L_N$-ClpB* and ΔN-ClpB* as model system, since ClpB hexamers dynamically exchange subunits ensuring stochastic formation of mixed hexamers (*Figure 5A/B*; *Haslberger et al., 2008*; *Werbeck et al., 2008*). We confirmed mixing of WT and mutant subunits by showing that the presence of ATPase-deficient ΔN-ClpB*-E218A/E618A, harboring mutated Walker B motifs in both AAA domains, strongly poisoned disaggregation activity of $L_N$-ClpB* (*Figure 5—figure supplement 1A*). Similarly, presence of an excess of ΔN-ClpB* inhibited $L_N$-ClpB* (*Figure 5—figure supplement 1B*). We determined the disaggregation activities of mixed $L_N$-ClpB*/ΔN-ClpB* hexamers formed at diverse mixing ratios. The determined activities were compared to those derived from a theoretical model assuming that a mixed hexamer only displays activity if it contains a certain number of $L_N$-ClpB* subunits (*Figure 5B*, *Figure 5—figure supplement 1C–E*). This comparison revealed that approx. four to five ClpL NTDs domains must be present in a hexamer to confer high disaggregation activity. In a reciprocal approach we tested whether an isolated NTD can inhibit $L_N$-ClpB* or ClpL disaggregation activity by competing for aggregate binding (*Figure 5C/D*). Addition of up to a 50-fold excess of ClpL NTD did not reduce disaggregation activities (*Figure 5C*: $p_{ANOVA}$ = 0.51; *Figure 5D*: $p_{ANOVA}$ = 0.18). This suggests a vast increase in binding affinities of AAA+ rings harboring multiple ClpL NTDs to protein aggregates and explains efficient outcompetition of isolated NTD by hexamers.

## Functional relevance of diverse ClpL assembly states

*S. pneumoniae* ClpL has been recently described as tetradecameric species, which forms by interaction of two heptameric rings via head-to-head interactions of coiled-coil MDs (*Kim et al., 2020*; *Figure 6A*). This assembly state is suggested to represent the active species of ClpL, however, it should render the NTDs largely inaccessible for aggregate binding. We therefore monitored *Lm* ClpL oligomerization by size exclusion chromatography (SEC) and negative staining EM. ClpL eluted prior to ClpB in SEC runs, indicating the formation of assemblies larger than hexamers (*Figure 6—figure supplement 1A*). EM analysis revealed that ATPγS-bound ClpL adopts diverse assembly states including single hexameric and heptameric rings, dimers of rings, and a tetrahedral structure formed by four ClpL rings (*Figure 6B*). A tetrahedral structure is also formed by the bacterial AAA+ member ClpC upon binding of activating compounds (*Maurer et al., 2019*; *Morreale et al., 2022*; *Taylor et al., 2022*). The structural plasticity of ClpL is therefore much larger than originally anticipated. Dimers and tetramers of ClpL rings form by MD-MD interactions as ClpL-ΔM (ΔD330-Q376) or MD mutants (E352A, F354A) (*Figure 6—figure supplement 1B*) only form single hexameric and heptameric rings and accordingly eluted much later in SEC runs as compared to ClpL-WT (*Figure 6C*, *Figure 6—figure supplement 1A/C*). Notably, ΔN-ClpL only forms dimers of rings, suggesting that NTD presence destabilizes this assembly state (*Figure 6C*, *Figure 6—figure supplement 1C*). An increased rigidity of ΔN-ClpL is also supported by a higher $T_M$-value as compared to ClpL-WT and ClpL-ΔM (*Figure 6—figure supplement 1D*). EM analysis of selected NTD point mutants revealed minor changes in the fractions of individual structural states as compared to ClpL-WT but did not resemble ΔN-ClpL (*Figure 6—figure supplement 1E*), suggesting that the mere presence of the NTD destabilizes a dimer of rings through steric clashes.

Disaggregation activities of ClpL MD mutants were either similar to WT (E352A, for luciferase: 101 ± 20% of WT activity; $p_{Luc}$ = 0.9; for MDH: 76 ± 1% of WT activity; $p_{MDH}$ = 7.3e-4) or partially reduced

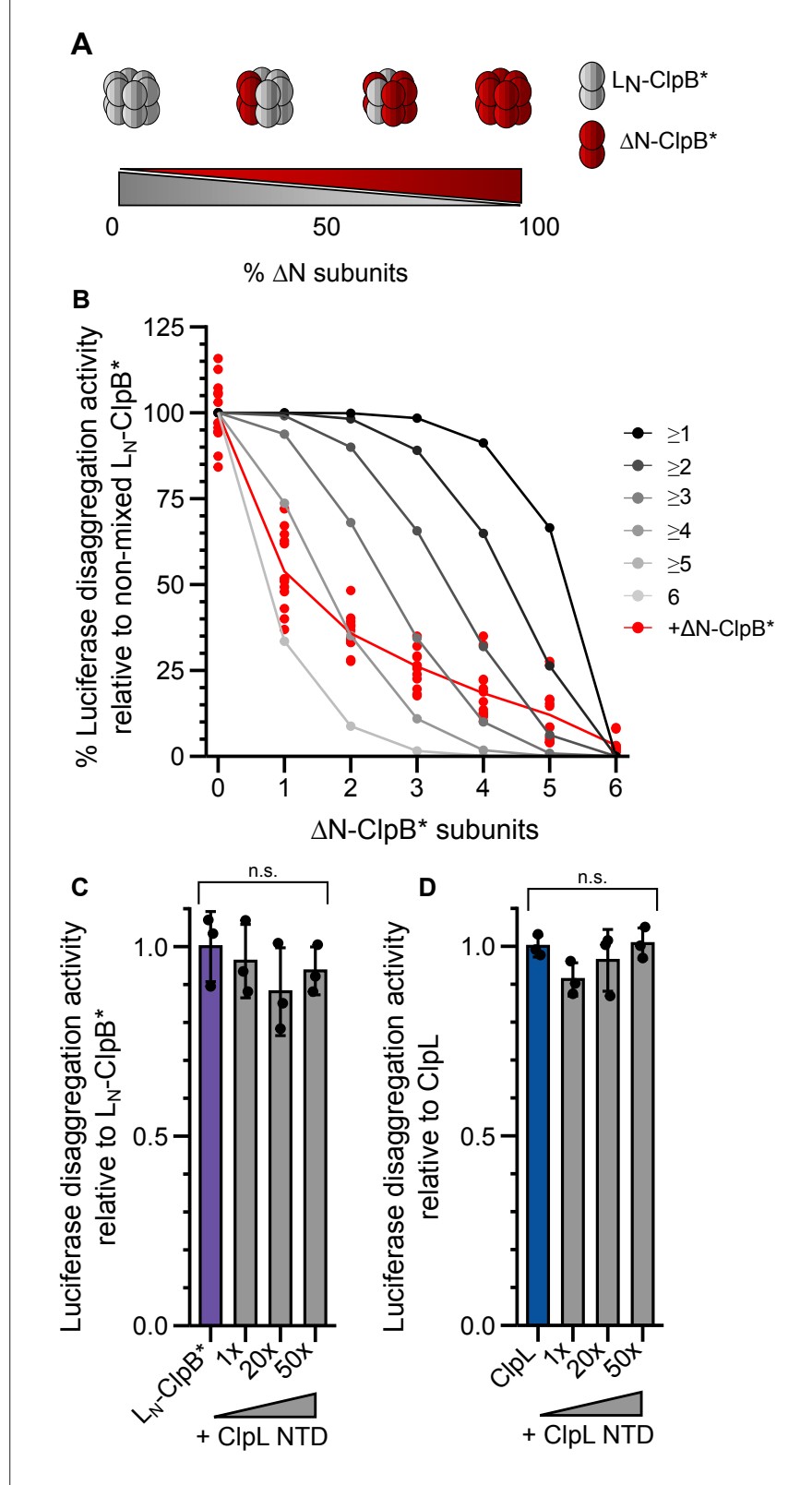

**Figure 5.** Multiple ClpL N-terminal domains (NTDs) are required for disaggregation activity. (**A**) Varying the ratio of $L_N$-ClpB* and ΔN-ClpB* leads to formation of mixed hexamers with diverse numbers of NTDs. (**B**) Luciferase disaggregation activities (% refolded luciferase/min) of mixed $L_N$-ClpB*/ΔN-ClpB* hexamers were determined and compared with curves calculated from a model (black to gray), which assumes that a mixed hexamer only displays

*Figure 5 continued on next page*

*Figure 5 continued*

disaggregation activity if it contains the number of NTDs indicated. Mixing ratios are indicated as number of ΔN-ClpB* in a hexamer. (C/D) Luciferase disaggregation activities of $L_N$-ClpB* (**C**) and ClpL (**D**) were determined in the absence and presence of an excess of isolated NTD as indicated. Disaggregation activities determined in NTD absence were set as 100%. Shown are data points (**B**) or data points and mean ± SD (C/D), n≥3. Statistical analysis: one-way ANOVA, Welch's test for post hoc multiple comparisons. Significance levels: *p<0.05; **p<0.01; ***p<0.001. n.s.: not significant.

The online version of this article includes the following figure supplement(s) for figure 5:

**Figure supplement 1.** Multiple ClpL N-terminal domains (NTDs) are required for disaggregation activity.

---

(F354A, for luciferase: 48 ± 3% of WT activity; $p_{Luc}$ = 1.6e-6; for MDH: 26 ± 9% of WT activity; $p_{MDH}$ = 1.8e-8) (*Figure 6D/E*). ClpL-F354A also exhibited a reduced ATPase activity (at 0.125 μM: 36 ± 7% of WT, p = 1.1e-5), particularly at low protein concentrations (*Figure 6—figure supplement 1F*). *Lg* ClpL, which is highly active in disaggregation assays (*Figure 1E*), forms hexamers as most abundant structural state (69%) (*Figure 6C*). We infer that single ClpL rings represent functional disaggregases, questioning a specific role of ring dimers in protein disaggregation. Our data also imply that ClpL activity is not restricted to a heptameric state as suggested before (*Kim et al., 2020*).

To study the role of dimers of rings in protein disaggregation, we generated the MD mutant ClpL-T355C, which allowed for disulfide bond formation between interacting MDs under oxidizing conditions (*Figure 6F*). This enabled us to probe for the consequences of stabilizing ClpL ring dimers through covalent linkages on disaggregation activity. T355C purified under reducing conditions (+DTT) exhibited reduced disaggregation (74 ± 23% of WT, p=0.012) and ATPase activities (at 0.125 μM: 73 ± 11% of WT, p=0.016) as compared to ClpL-WT (*Figure 6—figure supplement 2A/B*). Disulfide bond formation and breakage upon removal and re-addition of DTT was verified by SDS-PAGE (*Figure 6F*) and formation of ring dimers was confirmed by EM (*Figure 6—figure supplement 2C*). Luciferase disaggregation activities of ClpL-WT and ClpL-T355C were assessed in absence and presence of DTT. ClpL-WT, purified under reducing conditions, served as positive assay control, while oxidized ClpL-WT served as reference and treatment control for the purification under oxidizing conditions. The assay was in general sensitive to the presence of DTT, leading to an overall increase of luciferase reactivation by 2.7-fold (p=1.2e-8) for ClpL-WT, likely due to the fact that firefly luciferase itself possesses multiple cysteines. Importantly, we observed a much stronger activity gain of ClpL-T355C under reducing conditions as compared to ClpL-WT (2.8±0.4-fold [p=7.5e-10] vs 6.6±1.3-fold [p=5e-8], respectively) (*Figure 6G*). Re-addition of DTT strongly increased ClpL-T355C disaggregation activity, which was now similar to ClpL-WT. In an alternative approach, we monitored the unfolding activities of oxidized and reduced ClpL-WT and ClpL-T355C toward mixed luciferase-YFP aggregates (*Figure 6—figure supplement 2D*). Here, we followed the loss of YFP fluorescence as readout for disaggregation activity as we assumed that the unfolding reaction is more independent on DTT. Indeed, the activity of oxidized WT remained largely unaffected (1.4±0.5 fold [p=0.04] increase in activity with DTT). In striking contrast, ClpL-T355C showed massive loss in disaggregation activity in the absence of DTT but became fully active upon DTT re-addition (18.2±7.4 fold [p=2.4e-11] increase in activity with DTT). We infer that stabilizing ClpL ring dimers through covalent linkage strongly reduces disaggregation activity.

We finally asked for a potential physiological relevance of ClpL ring dimers and tetramers. We observed that ClpL MD mutants (E352A, F354A) are produced at lower levels in *Ec ΔclpB* cells as compared to ClpL-WT and ΔN-ClpL (*Figure 6—figure supplement 3A*). Reduced production levels of the mutants correlated with cellular toxicity at 42°C providing a potential rationale for their limited synthesis. ClpL-WT did not cause toxicity when produced at comparable levels, however higher production levels also exerted toxic effects (*Figure 6—figure supplement 3B*). Toxicity was dependent on the presence of the NTD as no toxicity was observed upon production of ΔN-ClpL. We infer that ClpL MD mutants, which only form single ring assemblies, create toxic effects in vivo. This points to protective roles of ClpL ring dimers and tetramers by restricting ClpL activity.

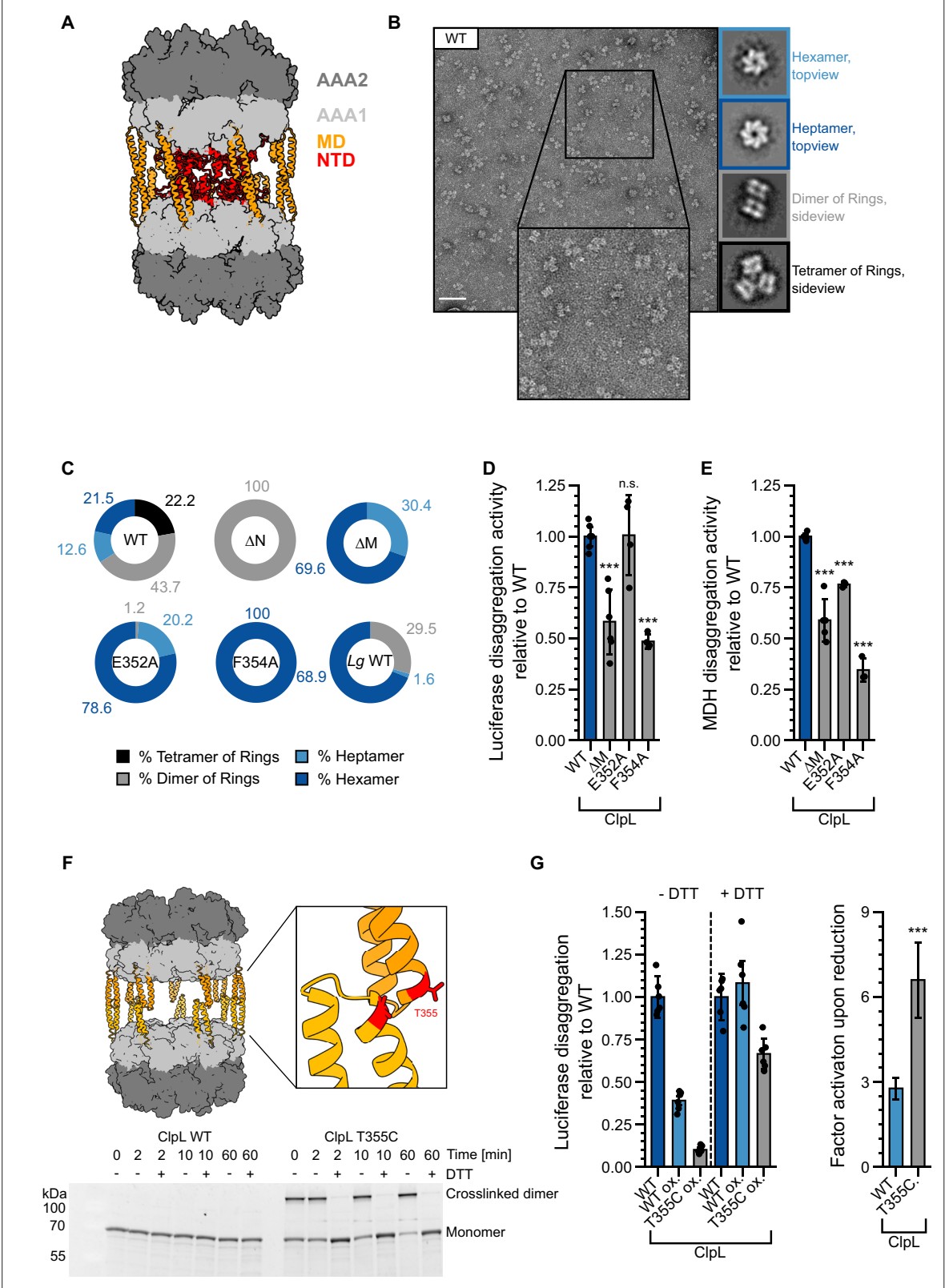

**Figure 6.** Stabilizing ClpL ring dimers strongly reduces disaggregation activity. (**A**) AlphaFold2 model of *L. monocytogenes* (*Lm*) ClpL ring dimers. Positions of individual domains are indicated. (**B**) Negative stain EM of *Lm* ClpL. Two-dimensional (2D) class averages revealing single ring hexamers or heptamers, ring dimers and tetramers of rings are indicated. The scale bar is 100 nm. (**C**) Populations (%) of diverse ClpL assembly states based on 2D class averages were determined for *Lm* ClpL wild-type (WT) and indicated mutants and *L. gasseri* (*Lg*) ClpL. Evaluated particles: $n_{WT}$ = 5233, $n_{\Delta N}$

*Figure 6 continued on next page*

*Figure 6 continued*

= 18,314, $n_{\Delta M}$ = 1900, $n_{E352A}$ = 14,215, $n_{F354A}$ = 7074, $n_{Lg\ WT}$ = 11,765. (D/E) Disaggregation activities of ClpL-WT and indicated M-domain (**M**) mutants toward aggregated luciferase (**D**) and malate dehydrogenase (MDH) (**E**) (each: % regain enzymatic activity/min) were determined. The activity of ClpL was set to 1. (**F**) Model of ClpL ring dimers. Positions of T355 residues in interacting M-domains are depicted. Crosslinking of ClpL-T355C was achieved by DTT removal and further incubation at room temperature (RT) as indicated. The formation of crosslinked ClpL-T355C dimers was monitored by SDS-PAGE and Coomassie staining. Addition of 10 mM DTT reversed the disulfide bonds. (**G**) Luciferase disaggregation activities of reduced ClpL-WT (assay control) and oxidized ClpL-WT (treatment control) or ClpL-T355C were determined in the absence of DTT (-DTT). Oxidized variants (WT and T355C) were additionally preincubated with 10 mM DTT for 30 min and tested for disaggregation activity in the presence of DTT (+DTT). The factor of increase in disaggregation activity upon reduction (+DTT) is indicated (right). Shown are data points and mean ± SD (D/E/G) or only mean ± SD (**G**), n≥3. Standard deviations for the activity gain factors (**G**) have been propagated from disaggregation activity standard deviations. Statistical analysis: one-way ANOVA, Welch's test for post hoc multiple comparisons. Significance levels: *p<0.05; **p<0.01; ***p<0.001. n.s.: not significant.

The online version of this article includes the following source data and figure supplement(s) for figure 6:

**Source data 1.** Source data includes the non-cropped and non-processed image of the SDS-gel.

**Source data 2.** Source data includes the non-cropped and non-processed image of the SDS-gel and indicates sections and loading schemata of the respective figure.

**Figure supplement 1.** ClpL rings interact in an M-domain-dependent manner.

**Figure supplement 1—source data 1.** Source data include non-cropped and non-processed images of SDS-gels.

**Figure supplement 1—source data 2.** Source data include non-cropped and non-processed images of SDS-gels and indicate sections and loading schemata of the respective figure supplement.

**Figure supplement 2.** Stabilizing ClpL ring dimers by disulfide crosslinking.

**Figure supplement 3.** Production of ClpL M-domain (MD) mutants cause increased toxicity in *E. coli*.

**Figure supplement 3—source data 1.** Source data includes the non-cropped and non-processed image of the SDS-gel.

**Figure supplement 3—source data 2.** Source data includes the non-cropped and non-processed image of the SDS-gel and indicates sections and loading schemata of the respective figure supplement.

## Discussion

In the presented work we describe the bacterial AAA+ member ClpL as potent and partner-independent disaggregase that confers enhanced heat resistance to *Lm*. The presence of ClpL on transmissible plasmids in *Lm* poses a severe risk on food safety as it substantially increases the survival rate of the pathogen in adverse environments like food processing plants. It also explains the high prevalence of the *clpL* gene in *Listeria* species isolated from ready-to-eat foods and dairy products (*Parra-Flores et al., 2021*; *Ramadan et al., 2023*).

ClpL shares various features with the recently described stand-alone ClpG disaggregase, which is linked to superior thermotolerance in selected Gram-negative bacteria, qualifying ClpL as ClpG counterpart in Gram-positive species. Our analysis of ClpL and previous findings on ClpG allow to define five features shared by both novel disaggregases.

First, ClpL and ClpG exhibit a higher threading power as compared to the canonical ClpB disaggregase. ClpB unfolding power is repressed by intersubunit interactions of its long coiled-coil MDs (*Deville et al., 2017*). The shortened lengths of ClpL and ClpG MDs explain why this inhibitory mechanism is not operative in the novel disaggregases. Increasing ClpB threading activity by MD mutations is linked to severe cellular toxicity, rationalizing ClpB limitations (*Lipińska et al., 2013*; *Oguchi et al., 2012*). The increased unfolding power of ClpL/ClpG is linked to higher disaggregation activities as shown for various aggregated model proteins in this work. We suggest that an enhanced threading activity will become particularly important upon extreme heat shock, which causes more complete unfolding of proteins. This will increase the number of interactions between proteins trapped inside an aggregate, thus demanding for an enhanced threading force for efficient disaggregation.

Second, ClpL and ClpG are more thermostable as compared to the DnaK/ClpB bi-chaperone disaggregase present in corresponding bacterial species. The $T_M$ value (46–50°C) of *Lm* DnaK might represent a limiting factor for *Lm* survival at temperatures above 46°C as loss of DnaK by thermal unfolding will abrogate DnaK/ClpB disaggregation activity. This will leave *Lm* cells unprotected against severe heat stress unless they harbor plasmid-encoded ClpL, which resists unfolding up to 60°C. Notably, unfolding of DnaK at high temperatures will abrogate the observed competition between DnaK and ClpL for aggregate binding, enabling for undisturbed activity of the more powerful disaggregase. Similarly, *Ec* ClpG is more heat stable than *Ec* DnaK ($T_M$ values of 69°C and 59°C, respectively)

(*Kamal et al., 2021a*; *Palleros et al., 1992*) and dramatically increases cell survival at 65°C. We infer that increased heat resistance provided by novel disaggregases is reflected in their enhanced thermostability.

Third, distinct NTDs enable novel disaggregases to directly bind to protein aggregates, making them independent of partner proteins. ClpL and ClpG NTDs do neither exhibit sequence nor structural homology, however, both use hydrophobic or aromatic residues for aggregate recognition (*Katikaridis et al., 2023*). The ClpL NTD only adopts a partially stable core structure with regions featuring aromatic residues being mobile and accessible, suggesting that it represents a sticky tentacle that interacts via aromatic residues with hydrophobic patches present on an aggregate surface. This binding mode is reminiscent of small heat shock proteins, which interact via disordered NTDs enriched in aromatic residues with substrates (*Kriehuber et al., 2010*; *Shrivastava et al., 2022*). Both, ClpL and ClpG, require simultaneous binding of multiple NTDs to protein aggregates for high disaggregation activities (*Katikaridis et al., 2023*), indicating that they increase binding affinity through avidity effects. This principle provides substrate specificity while sparing non-aggregated protein species from potentially harmful threading and unfolding activities.

Fourth, novel disaggregases lack the specific IGL/F signature motif, which is essential for cooperation of other Hsp100 proteins with the peptidase ClpP. This feature is shared with the canonical ClpB disaggregase (*Weibezahn et al., 2004*), suggesting that protein disaggregation is primarily linked to protein refolding. This also underlines that it is the loss of essential proteins upon heat-induced aggregation that represents the major limiting factor for bacterial viability.

Fifth, novel disaggregases are typically encoded on transmissible plasmids or mobile genomic islands. ClpL present in other Gram-positive bacteria is either encoded on plasmids (*Huang et al., 1993*) or is part of the core genome and can be flanked by inverted repeat sequences and truncated transposase genes, suggesting acquisition via horizontal gene transfer (*Suokko et al., 2005*). Notably, prolonged cultivation at high temperatures can lead to mobilization of chromosomal-encoded *clpL* (*Suokko et al., 2005*). We infer that novel disaggregases can be acquired by horizontal gene transfer, leading to spreading of bacterial resistance toward severe heat stress in Gram-positive and Gram-negative bacteria.

*Lm* ClpL displays remarkable structural plasticity by forming hexameric and heptameric rings that can interact via MDs forming higher assembly states. This confirms and expands a former report on *S. pneumoniae* ClpL (*Kim et al., 2020*). ClpL MD mutants, which only form single hexameric rings, remain functional disaggregases and, similarly, the disaggregase *Lg* ClpL mostly forms single hexamers. This suggests that single rings of ClpL represent the active species. It is worth noting that MD mutants F354A and, to a lesser extent, T355C, exhibited partial reduction in disaggregation and ATPase activities, suggesting an auxiliary function of the MD in controlling these activities. The aggregate-targeting NTDs will be obstructed in dimers and tetramers of rings precluding ClpL binding to an aggregated protein. Accordingly, stabilizing ring dimers via covalent linkage leads to a loss of ClpL disaggregation activity. The finding that ΔN-ClpL exclusively forms dimers of rings implies that NTDs destabilize such structural states potentially by imposing a steric hindrance on MD-MD interactions. The NTDs may thereby increase ring dynamics, ensuring their accessibility.

The formation of ring dimers is also observed for other AAA+ family members and their functional roles are currently actively discussed. p97 double rings form by ATPase ring interactions and are suggested to represent a potentiated state (*Gao et al., 2022*). Dodecamers of the bacterial AAA+ protease Lon regulate substrate selectivity and mostly exhibit reduced proteolytic activity (*Vieux et al., 2013*). The role of ring dimers of the human mitochondrial disaggregase Skd3 in protein disaggregation is controversial and both higher and lower disaggregation activity has been associated with this state (*Cupo et al., 2022*; *Gupta et al., 2023*; *Lee et al., 2023*; *Spaulding et al., 2022*; *Wu et al., 2023*). The mechanistic role of ClpL ring assemblies remains to be explored. We speculate that they might represent storage states, reducing ClpL activity by shielding NTDs. We observed toxicity upon production of ClpL MD mutants in *Ec* cells, pointing to a physiological role of ClpL ring assemblies in controlling ClpL activity. Studying the dynamics and regulations of ClpL ring interactions will become crucial for understanding ClpL activity control.

## Materials and methods

### Strains and plasmids

All strains and plasmids used in this study are summarized in *Supplementary file 1 Ec* cells were grown in LB medium at 30°C containing appropriate antibiotics (kanamycin [50 µg/ml], ampicillin [100 µg/ml], and spectinomycin [50 µg/ml]), shaking at 120 rpm. Deletion mutants and $L_N$-ClpB* hybrids were generated by PCR/restriction cloning, point mutants were constructed via PCR-based mutagenesis. ClpL NTD mutants A-E were generated utilizing synthetic DNA as template (Invitrogen) and restriction cloning. All constructs were verified by sequencing.

### Protein purification

*Ec* ClpB (WT and derivatives) was purified after overproduction from *Ec* Δ*clpB::kan* cells using pDS56-derived expression vectors (*Oguchi et al., 2012*). *Pa* ClpG$_{GI}$, *Lm* ClpL (WT, NTD mutants and isolated NTD), *Lg* ClpL and components of the *L. m* BKJE system were purified after overproduction in *Ec* BL21 cells using pET24a-derived expression vectors. $^{15}N$ single-labeled and $^{15}N$-$^{13}C$ double-labeled ClpL NTDs for NMR were produced in *Ec* BL21 cells in M9 medium with 1 g/l $^{15}NH_4Cl$ and 2 g/l $^{13}C$-glucose. *Lm* ClpL MD mutants and $L_N$-ClpB* fusion constructs were overexpressed in *Ec* BL21 cells from pCA528 and pC6AmpΔBsaI expression vectors, fusing an Ulp1 cleavable 6x His+SUMO-tag to the N-termini, protecting the ClpL N-termini from cleavage upon cell lysis and allowing for selective purification of ClpL variants with intact NTDs.

All proteins were purified using Ni-IDA (Macherey-Nagel) following the instructions provided by the manufacturer. In short, cell pellets were resuspended in buffer A1 (50 mM NaH$_2$PO$_4$, 300 mM NaCl, 5 mM β-mercaptoethanol [β-Me], pH 8.0) supplemented with protease inhibitors (8 µg/ml Pepstatin A, 10 µg/ml Aprotinin, 5 µg/ml Leupeptin) and DnaseI (5 µg/ml). For ClpL purifications a higher salt concentration was used in buffer A1* (50 mM NaH$_2$PO$_4$ pH 8.0, 1 mM NaCl, 5 mM β-Me), to reduce co-purification of partially cleaved monomers in assembled oligomers. After cell lysis using a French press the cell debris was removed by centrifugation at 17,000×*g* for 1 hr at 4°C and the cleared lysate was incubated with Protino IDA resin (Macherey-Nagel) for 20 min at 4°C. Afterward the resin was transferred into a plastic column and washed with at least five CV buffer A1, or in the case of ClpL, washed with at least five CV buffer A1* and then five CV buffer A1. His-tagged proteins were eluted by addition of buffer A1 supplemented with 250 mM imidazole. For *Lm* DnaJ purification cell pellets were lysed in lysis buffer J1 (50 mM Tris pH 7.9, 500 mM NaCl, 0,6% Brij58, 5 mM MgCl$_2$ 1× DNAseI, cOmplete Mini EDTA-free protease inhibitor tablets [Roche]). The resin was first washed with at least 10 CV of wash buffer J2 (50 mM Tris pH 7.9, 1.5 M NaCl, 0.1% Brij58, 2 M urea) and then with 20 CV of wash buffer J3 (50 mM Tris pH 7.9, 500 mM NaCl, 2 mM β-mercaptoethanol, 2 M urea) to remove detergent. DnaJ was eluted with elution buffer J4 (50 mM Tris pH 7.9, 500 mM NaCl, 2 mM β-mercaptoethanol, 2 M urea) and imidiazole was removed via overnight dialysis in storage buffer J5 (40 mM HEPES pH 7.6, 2 mM β-mercaptoethanol, 300 mM KCl, 10% glycerol [vol/vol]) at 4°C.

Except for DnaJ and the ClpL all NQ/Ala mutant all proteins were further purified via SEC. Prior to that SUMO-tagged proteins were incubated with Ulp1 for 1 hr at room temperature (RT) to cleave off the His$_6$-SUMO tag. SEC runs were done with HiLoad 16/600 Superdex 200 pg (Cytiva) in buffer B1 (50 mM Tris pH 7.5, 50 mM KCl, 10 mM MgCl$_2$, 5% [vol/vol] glycerol, 2 mM DTT) for *Ec* ClpB or buffer B2 (50 mM Tris pH 7.5, 50 mM KCl, 20 mM MgCl$_2$, 5% [vol/vol] glycerol, 2 mM DTT) for all others. In case of isolated ClpL NTD a HiLoad 16/600 Superdex 30 pg (GE Healthcare) was used in the SEC step. Proteins were concentrated with either Aquacide II (Merck) or Amicon Ultra Centrifugal Filters. Isolated ClpL NTD was dialyzed in NMR buffer (50 mM NaH$_2$PO$_4$ pH 6.5, 50 mM KCl, 2 mM DTT) overnight at 4°C.

Purifications of *Ec* DnaK, DnaJ, GrpE, and firefly luciferase were performed as described previously (*Haslberger et al., 2008*; *Oguchi et al., 2012*; *Seyffer et al., 2012*). Pyruvate kinase of rabbit muscle and MDH of pig heart muscle were purchased from Sigma. Protein concentrations were determined with via Bradford assay (Bio-Rad Protein Assay Dye Reagent).

## Biochemical assays

### ATPase assay

The ATPase activities of 0.125 or 1 µM ClpL, ClpL mutants, and $L_N$-ClpB* were determined in a reaction volume of 100 µl in assay buffer (50 mM Tris pH 7.5, 50 mM KCl, 20 mM MgCl$_2$, 2 mM DTT) with 0.5 mM NADH (Sigma), 1 mM PEP (Sigma), and 1/100 (vol/vol) Pyruvate Kinase/Lactic Dehydrogenase mix (Sigma). 100 µl of 4 mM ATP in assay buffer (50 mM Tris pH 7.5, 50 mM KCl, 20 mM MgCl$_2$, 2 mM DTT) were added to each reaction in a 96-well plate (TPP) format to start the reaction. The decrease of NADH absorbance at 340 nm was determined in a CLARIOstar Plus Platereader (BMG Labtech) at 30°C. ATPase activities were calculated assuming a 1:1 stoichiometry of NAD$^+$ oxidation and the production of ADP.

### In vitro disaggregation assays

All disaggregation assays were performed in 50 mM Tris pH 7.5, 50 mM KCl, 20 mM MgCl$_2$, 2 mM DTT. Aggregates of 200 nM luciferase were generated through incubation at 46°C for 15 min in assay buffer. Aggregates of 2 µM MDH were generated through incubation at 47°C for 30 min. Aggregates of 1 µM GFP were generated through incubation at 80°C for 10 min. Aggregates of 2 µM α-glucosidase were generated through incubation at 47°C for 30 min. Equal volumes of protein aggregate suspensions were mixed with chaperones and an ATP regeneration system (15 mM PEP, 20 ng/µl pyruvate kinase). Reactions were started by addition of 2 mM ATP. If not stated otherwise final chaperone concentrations were: 1 µM ClpL, ClpG, ClpB, $L_N$-ClpB*, *Ec* KJE: 1 µM *Ec* DnaK, 0.2 µM *Ec* DnaJ, 0.1 µM *Ec* GrpE, *Lm* KJE: 1 µM *Lm* DnaK 0.5 µM *Lm* DnaJ, 0.25 µM *Lm* GrpE, 2× Lm KJE: 2 µM *Lm* DnaK 1 µM *Lm* DnaJ, 0.5 µM *Lm* GrpE. Reactions were kept at 30°C for the duration of the assay.

To determine luciferase activities 2 µl were taken from each disaggregation reaction in regular time intervals and added to 100 µl measurement buffer (25 mM glycylglycine pH = 7.4, 12.5 mM MgSO$_4$, 5 mM ATP) inside a 5 ml reaction tube (Greiner). Luminescence was measured for 2 s in a Lumat LB 9507 (Berthold Technologies) after injection of 100 µl 250 µM luciferin (Gold Biotechnology). Luciferase refolding served as a measure for disaggregation activity, native luciferase was used as a reference for the efficiency of recovery. Refolding rates (% luciferase refolded/min) were calculated from the linear increase in luciferase activities.

MDH refolding was measured in additional presence of 1 µM GroEL/ES to facilitate refolding of disaggregated MDH. 10 µl were taken from each disaggregation reaction in regular time intervals and mixed with 690 µl measurement buffer (150 mM potassium phosphate pH 7.6, 0.5 mM oxaloacetate, 0.28 mM NADH, 2 mM DTT) inside a 1 ml polystyrene cuvette (Sarstedt). MDH activity was quantified by measuring absorption at 340 nm every 10 s for 30 s on a Biochrom Novaspec Plus photometer. The activity of native MDH served as a reference to determine disaggregation efficiency. Refolding rates (% refolded MDH/min) were calculated from the linear increase in MDH activity.

Refolding of aggregated GFP was determined by monitoring regain of GFP fluorescence upon excitation at 400 nm was continuously recorded at 510 nm on an LS50B spectrofluorometer (PerkinElmer).

Disaggregation of α-glucosidase was followed by continuously recording aggregate turbidity using 600 nm as excitation and emission wavelengths on a LS50B spectrofluorometer (PerkinElmer).

### Luciferase-YFP-unfolding assay

200 nM luciferase-YFP (Luc-YFP) was subjected to heat treatment at 46°C for 15 min. 150 µl aggregate suspension was combined with the respective disaggregases. Reactions were started and sustained by addition of 2 mM ATP and an ATP regeneration system (15 mM PEP, 20 ng/µl pyruvate kinase) in a final reaction volume of 300 µl. Chaperone concentrations were 1 µM ClpL, ClpG, ClpB, *Ec*/*Lm* DnaK, 0.2 µM *Ec* DnaJ, 0.5 µM *Lm* DnaJ, 0.1 µM *Ec* GrpE, (0.25 µM *Lm* GrpE). Reactions were kept in 1 ml UV quartz cuvettes at 30°C for the duration of the assay. YFP fluorescence upon excitation at 505 nm was continuously recorded at 525 nm on a LS50B spectrofluorometer (PerkinElmer). Alternatively, fluorescence was tracked using a CLARIOstar Plus Platereader (BMG Labtech). Here, the samples were scaled down to a total volume of 20 µl and loaded onto Black Polysterene Non-Binding 384 Assay Plates (Corning, flat bottom). The platereader assay was performed at RT, due to positional bias of the heating function. The initial signal was set to 100% for each sample. Unfolding rates were calculated from the linear phase of YFP fluorescence decrease.

## Protein aggregate binding assay

To examine the interaction between ClpL or $L_N$-ClpB* with protein aggregates, 4 µM MDH was heat-denatured at 47°C for 30 min in assay buffer (50 mM Tris pH 7.5, 50 mM KCl, 20 mM $MgCl_2$, 2 mM DTT). MDH aggregates were mixed with 1.5 µM ClpL and 2 mM of ATPγS in 100 µl assay buffer and incubated at 25°C for 10 min. Soluble and insoluble fractions were separated by centrifugation at 13,000 rpm for 25 min at 4°C. The pellet fraction was washed once with 150 µl assay buffer and centrifuged again at 13,000 rpm for 10 min at 4°C. Binding assays were performed in low binding microtubes (Sarstedt). Supernatant and pellet fractions were mixed with protein sample buffer and analyzed by Coomassie staining after SDS-PAGE. As a control, ClpL or $L_N$-ClpB* without aggregated proteins was subjected to the same protocol. Band intensities of supernatant and pellet fractions were quantified using ImageJ and the percentage of chaperone in the pellet fraction, reflecting binding to protein aggregates, was determined. Background levels of chaperones in the pellet fractions in absence of protein aggregates were subtracted. In case of aggregate binding competition experiments 1.5 µM ClpL-E197A/E530A was co-incubated with the *Lm* or *Ec* DnaK chaperone system (1/0.5 µM *Lm* DnaK/DnaJ; 1/0.2 µM *Ec* DnaK/DnaJ).

## Determination of $T_M$ values by SYPRO Orange binding and nanoDSF

Thermal stability of protein samples was monitored in a thermal shift assay with SYPRO Orange (Thermo Fisher Scientific) on a LightCycler 480 II (Roche). All proteins were dialyzed and stored in HEPES storage buffer (50 mM HEPES pH 7.5, 50 mM KCl, 20 mM $MgCl_2$, 5% glycerol) to reduce thermally induced pH changes.

Chaperones were incubated at concentration of 1 µM with 2 mM of the respective nucleotide (ATPγS/ATP/ADP) in HEPES low salt buffer (50 mM HEPES pH 7.5, 50 mM KCl, 20 mM $MgCl_2$) in a total volume of 50 µl for 1 hr (ATPγS) or 5 min (ATP/ADP) to allow for the formation of a stable distribution of oligomeric species. Subsequently, SYPRO Orange (from 5000× commercial stock) was added to a final concentration of 80× and the samples were loaded onto a LightCycler 480 Multiwell Plate 384. SYPRO Orange Fluorescence was recorded over a temperature range from 20°C to 95°C. NanoDSF (differential scanning fluorimetry) measurements were performed on a Prometheus Panta from NanoTemper Technologies GmbH, using Panta Control v1.4.3 software. Chaperone and nucleotide concentrations were the same as for the SYPRO experiments, only for DnaK a chaperone concentration of 10 µM was used, due to insufficient signal at lower concentrations. $T_M$-values were derived from inflection points based on sigmoidal curve fits (https://gestwickilab.shinyapps.io/dsfworld/) for SYPRO Orange data, while $T_M$-values for NanoDSF were directly calculated with the NanoTemper Analysis Software (v1.4.3).

## Disulfide crosslinking

For the purification of oxidized, disulfide bond-stabilized ClpL-T355C-tetradecamers and oxidized ClpL-WT, 5 mM β-Me was present in all steps up to the SEC, to prevent unspecific disulfide bond formation. 2 mM ATP was added during the elution step from the Ni-IDA resin and to the SEC running buffer (50 mM Tris pH 7.5, 50 mM KCl, 20 mM $MgCl_2$, 10% glycerol) to allow for efficient and correct ring assembly. No reducing agent was added during any of the later steps, while remaining β-Me was removed during the SEC. The samples were processed immediately and freeze/thaw steps of the reducing agent-free samples were avoided. Addition of oxidizing compounds (e.g. $Cu^{2+}$-phenanthroline) only slightly enhanced disulfide bond formation but also led to loss of ClpL-WT activity and was therefore avoided. The efficiency of the crosslinking was determined via SDS-PAGE, by comparing the ratio between ClpL dimers and monomers, whereas the overall quality of the assembly was assessed via negative stain EM. ClpL-T355C samples that showed a crosslinking efficiency of approx. 75% or higher were analyzed in two activity assays, with and without prior incubation of the chaperone with 10 mM DTT for 30 min at RT. (1) First, in a variant of the luciferase refolding assay, in which the assay buffer included 5 µM β-Me instead of 2 mM DTT, increasing luciferase refolding without breaking disulfide bonds. (2) Second, in the luciferase-YFP-unfolding assay, which was conducted without any further addition of reducing agent, except for the DTT added to indicated samples.

In both experiments ClpL-WT, purified under reducing conditions (see Protein purification), served as positive control for the assay, while oxidized ClpL-WT served as reference and treatment control for oxidized ClpL-T355C.

## Analytic SEC

Oligomerization of ClpL-WT and mutants was investigated by SEC using a Superose 6 10/300 GL column (GE Healthcare) at 4°C in assay buffer supplemented with 2 mM ATP. 6 µM protein was incubated for 5 min at RT with ATPγS before injection. Elution fractions were analyzed via SDS-PAGE followed by staining with SYPRO Ruby (Invitrogen) according to the manufacturer's protocols. Bio-Rad Gel Filtration Standard served as a reference.

## Bioinformatic analysis

Multiple sequence alignments were performed using Clustal Omega (https://www.ebi.ac.uk/Tools/msa/clustalo/) and were displayed using Jalview. An incomplete three-dimensional (3D) model of *Lm* ClpL was generated by SWISS-MODELL (https://swissmodel.expasy.org) using pdb-file 6LT4 as template (*S. pneumoniae* ClpL).

## Heat resistance assay

*Ec ΔclpB* or *dnaK103* cells harboring pUHE21/pDS56 derivates allowing for IPTG-controlled expression of *E. c. clpB*, *dnaK*, P.a. *clpG*, *Lm clpL*, and *ΔN-clpL* were grown in LB media at 30°C to early logarithmic phase (OD$_{600}$: 0.15–0.2). Expression of the respective proteins was induced by addition of IPTG (*clpB*: 10 µM, *dnaK*: 100 µM *clpG*: 100 µM, *clpL*: 100–500 µM). Protein production was documented 2 hr after IPTG addition by SDS-PAGE and western blot analysis using ClpL-specific antibodies. Subsequently 1 ml aliquots were shifted to 49°C for 120 min. At indicated time points bacterial survival was determined by preparing serial dilutions, spotting them on LB plates followed by incubation for 24 hr at 30°C.

## In vivo luciferase disaggregation assay

For in vivo luciferase disaggregation *Ec ΔclpB* cells harboring *placIq-luciferase* (for constitutive expression of luciferase) and either *pUHE21-clpB*, *pUHE21-clpG*, *pDS56-clpL*, or *pUHE21-ΔN-clpL* were grown in LB medium supplemented with ampicillin (100 µg/ml) and spectinomycin (50 µg/ml) at 30°C to early-logarithmic growth phase. *Chaperone* expression was induced by addition of IPTG (10 µM ClpB, 100 µM ClpG, 100/500 µM for ClpL-WT, or ΔN-ClpL) for 2 hr. Production of chaperones to similar levels was verified by SDS-PAGE and subsequent Coomassie staining. Luciferase activities were determined in a Lumat LB 9507 and set to 100%. For that, 100 µl of cells were transferred into 5 ml reaction tubes (Greiner), 100 µl of 250 µM luciferin were injected and luminescence was measured for 10 s. Next, 900 µl aliquots of cells were shifted to 46°C for 15 min. Immediately afterward, tetracycline (70 µg/ml) was added to stop protein synthesis and cells were moved back to 30°C. Luciferase activities were determined in regular intervals for 2 hr.

## Western blotting

Total extracts of cells were prepared and separated by SDS-PAGE, which was subsequently electrotransferred onto a PVDF membrane. The membrane was incubated in the blocking solution (3% bovine serum albumin [wt/vol] in TBS) for 1 hr at RT. Protein levels were determined by incubating the membrane with ClpL-specific antibodies (1:10,000 in TBS-T+3% [wt/vol] bovine serum albumin) and an anti-rabbit alkaline phosphatase conjugate (Vector Laboratories) as the secondary antibody (1:20,000). Blots were developed using ECF Substrate (GE Healthcare) as the reagent and imaged via Image-Reader LAS-4000 (Fujifilm). Band intensities were quantified with ImageJ.

## Electron microscopy and image processing

Negative staining, data collection, and processing were performed as described previously (*Gasse et al., 2015*). 1 µM ClpL samples were preincubated in the presence of 2 mM ATPγS for 5 min at RT and diluted immediately before application to a final concentration of 250 nM (except ClpL E352A and F354A [200 nM] and *Lg* ClpL [400 nM]). 5 µl sample was applied to a glow-discharged grid covered with an approximately 6–8 nm thick layer of continuous carbon. After incubation for 5 s, the sample was blotted on a Whatman filter paper 50 (1450-070) and washed with three drops of water. Samples on grids were stained with 3% aqueous uranyl acetate. Images were acquired on a Thermo Fisher Talos L120C electron microscope equipped with a Ceta 16M camera, operated at 120 kV. The micrographs were acquired at ×57,000 magnification (resulting in 2.26 Å per pixel) using EPU software. For

2D classification 20k particles (for *Lg* ClpL 25k) were selected using the boxer in EMAN2 (*Tang et al., 1998*). Image processing was carried out using the IMAGIC-4D package (*van Heel et al., 1996*). Particles were band-pass filtered, normalized in their gray value distribution, and mass centered. 2D alignment, classification, and iterative refinement of class averages were performed as previously described (*Liu and Wang, 2011*). For quantification only four unambiguously identifiable types of classes were considered: ClpL hexameric-ring top views, ClpL heptameric-ring top views, ClpL double-ring side views, and ClpL tetramers-of-rings. Classes of partial assemblies, contaminants, and picking-related artifacts were not included in the quantification. Number of quantification relevant particles: ClpL-WT = 5233, ClpL ΔN=18,314, ClpL ΔM=1900, ClpL E352A=14,215, ClpL F354A=7074, ClpL AB = 3588, ClpL C=7404, ClpL all Aro/Ala = 9655, *Lg* ClpL = 11,765.

## Nuclear NMR

All NMR experiments were recorded at 298 K using Bruker Avance III NMR spectrometers operating at magnetic field strengths, corresponding to $^{1}$H Larmor frequencies of 600, 700, and 800 MHz. The 600 and 800 MHz spectrometers were equipped with a cryogenic triple-resonance probe (the 700 MHz spectrometer was equipped with an RT triple-resonance probe). Protein concentrations were in the range between 200 and 620 µM. Backbone and side chain assignments were obtained using standard $^{1}$H,$^{13}$C,$^{15}$N scalar correlation experiments (*Sattler et al., 1999*) employing apodization weighted sampling (*Simon and Köstler, 2019*). The data were processed with NMRpipe (*Delaglio et al., 1995*) and analyzed with Cara (http://cara.nmr.ch) and Sparky (*Lee et al., 2015*). Secondary structure determination based on secondary chemical shifts (Cβ and Cα) was performed according to the study by *Wishart and Sykes, 1994*. NOEs for validation of AlphaFold2 structure predictions were obtained from $^{1}$H,$^{1}$H-2D NOESY (in $D_2O$), $^{15}$N- and $^{13}$C-edited 3D NOESY-HSQC, and $^{13}$C-edited 3D HMQC-NOESY (in $D_2O$) experiments. NOE assignments were performed manually using Cara.

$^{15}$N spin relaxation parameters $R_1$, $R_2$, and heteronuclear NOEs were recorded on the 600 MHz Avance III spectrometer at 298 K. For $R_1$ experiments, relaxation delays of 50, 100, 150, 200, 300, 400, 500, 700, and 1000 were used, where the 50, 500, and 700 ms delay were acquired in duplicates to estimate peak volume uncertainties. For $R_2$ experiments, relaxation delays of 16, 32, 64, 96, 112, 128, 160, 192, 224, and 256 ms were used, where the 16 and 64 ms delays were acquired in duplicates to estimate peak volume uncertainties. For $^{15}$N spin relaxation data analysis (peak volume, exponential fitting, and calculation of relaxation rates) the program PINT was used (*Ahlner et al., 2013*; *Niklasson et al., 2017*).

## Statistical analysis

We tested data from all relevant sets for significance via one-way ANOVA, where significant cases were followed by post hoc multiple comparisons with Welch's test, as all measurements were independent and both variance and sample size were not generally equal. The exact number of replicates for each data set is listed in *Supplementary file 2*. If not otherwise indicated, pairwise comparisons were made between the reference sample (no significance indicator) and individual test samples. For our most frequently used assays (luciferase refolding, MDH refolding, and ATPase assay) we chose to show and compare data relative to fixed controls, to reduce the impact of unspecific variance between repetitions. Real activities of the references are shown in separate plots.

# Acknowledgements

VB and PK were supported by the Heidelberg Biosciences International Graduate School (HBIGS). This work was supported by a grant of the Deutsche Forschungsgemeinschaft (MO970/7-1) to AM. We acknowledge access to the infrastructure of the Cryo-EM Network at the Heidelberg University (HDcryoNET). JH gratefully acknowledges the European Molecular Biology Laboratory (EMBL) for support. This work was supported by a grant of the Deutsche Forschungsgemeinschaft (MO970/7-1) to AM.

## Additional information

### Funding

| Funder | Grant reference number | Author |
|---|---|---|
| Deutsche Forschungsgemeinschaft | MO970/7-1 | Axel Mogk |

The funders had no role in study design, data collection and interpretation, or the decision to submit the work for publication.

### Author contributions

Valentin Bohl, Bernd Simon, Data curation, Formal analysis, Visualization, Methodology; Nele Merret Hollmann, Dirk Flemming, Data curation, Formal analysis, Methodology; Tobias Melzer, Panagiotis Katikaridis, Data curation, Formal analysis; Lena Meins, Data curation; Irmgard Sinning, Methodology; Janosch Hennig, Formal analysis, Visualization, Methodology, Writing – review and editing; Axel Mogk, Conceptualization, Data curation, Formal analysis, Funding acquisition, Writing - original draft, Project administration

### Author ORCIDs

Irmgard Sinning ⓘ http://orcid.org/0000-0001-9127-4477
Janosch Hennig ⓘ https://orcid.org/0000-0001-5214-7002
Axel Mogk ⓘ http://orcid.org/0000-0003-3674-5410

Reviewer #1 (Public Review): https://doi.org/10.7554/eLife.92746.3.sa1
Reviewer #2 (Public Review): https://doi.org/10.7554/eLife.92746.3.sa2
Reviewer #3 (Public Review): https://doi.org/10.7554/eLife.92746.3.sa3
Author response https://doi.org/10.7554/eLife.92746.3.sa4

## Additional files

### Supplementary files

- MDAR checklist

- Supplementary file 1. *E. coli* strains and plasmids used in this study.

- Supplementary file 2. Number of replicates.

### Data availability

All data are contained within the manuscript. The chemical shift assignments of ClpL NTD have been deposited at BMRB (accession code 52068).

The following dataset was generated:

| Author(s) | Year | Dataset title | Dataset URL | Database and Identifier |
|---|---|---|---|---|
| Hennig J | 2023 | ClpL NTD | https://bmrb.io/data_library/summary/index.php?bmrbId=52068 | BMRB, 52068 |

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
