## [Editor Report · eLife assessment]

This **important** manuscript details the characterization of ClpL from *L. monocytogenes* as an effective and autonomous AAA+ disaggregase that provides enhanced heat resistance to this food-borne pathogen. Supported by **compelling** evidence, the authors demonstrate that ClpL has DnaK-independent disaggregase activity towards a variety of aggregated model substrates, which is more potent than that observed with the endogenous canonical DnaK/ClpB bi-chaperone system. The work will be of broad interest to microbiologists and biochemists.

---

## [Referee Report · Reviewer #1 (Public Review)]

Summary:

This work describes the mechanism of protein disaggregation by the ClpL AAA+ protein of Listeria monocytogenes. Using several model subtrate proteins the authors first show that ClpL possesses a robust disaggregase activity that does not further require the endogenous DnaK chaperone in vitro. In addition, they found that ClpL is more thermostable than the endogenous L. monocytogenes DnaK and has the capacity to unfold tightly folded protein domains. The mechanistic basis for the robust disaggregase activity of ClpL was also dissected in vitro and in some cases, supported by in vivo data performed in chaperone-deficient *E. coli* strains. The data presented show that the two AAA domains, the pore-2 site and the N-terminal domain (NTD) of ClpL are critical for its disaggregase activity. Remarkably, grafting the NTD of ClpL to ClpB converted ClpB into an autonomous disaggregase, highlighting the importance of such a domain in the DnaK-independent disaggregation of proteins. The role of the ClpL NTD domain was further dissected, identifying key residues and positions necessary for aggregates recognition and disaggregation. Finally, using sets of SEC and negative staining EM experiments combined with conditional covalent linkages and disaggregation assays the authors found that ClpL shows significant structural plasticity, forming dynamic hexameric and heptameric active single rings that can further form higher assembly states via their middle domains.

Strengths:

The manuscript is well written and the experimental work well executed. It contains a robust and complete set of in vitro data that push further our knowledge of such important disaggregases. It shows the importance of the atypical ClpL N-terminal domain in the disaggregation process as well as the structural malleability of such AAA+ proteins. More generally, this work expands our knowledge of heat resistance in bacterial pathogens.

Weaknesses:

There is no specific weakness in this work, although it would have helped to have a drawing model showing how ClpL performs protein disaggregation based on their new findings. The function of the higher assembly states of ClpL remains unresolved and will need further extensive research. Similarly, it will be interesting in the future to see whether the sole function of the plasmid encoded ClpL is to cope with general protein aggregates under heat stress.

---

## [Referee Report · Reviewer #2 (Public Review)]

The manuscript by Bohl et al. is an interesting and carefully done study on the biochemical properties and mode of action of potent autonomous AAA+ disaggregase ClpL from Listeria monocytogenes. ClpL is encoded on plasmids. It shows high thermal stability and provides Listeria monocytogenes food-pathogen substantial increase in resistance to heat. The authors show that ClpL interacts with aggregated proteins through the aromatic residues present in its N-terminal domain and subsequently unfolds proteins from aggregates translocating polypeptide chains through the central pore in its oligomeric ring structure. The structure of ClpL oligomers was also investigated in the manuscript. The results suggest that mono-ring structure and not dimer or tetramer of rings, observed in addition to mono-ring structures under EM, is an active specie of disaggregase. In the revised version additional data is presented suggesting that dimer or tetramer of ClpL rings play a protective role in cell by restricting ClpL activity.

Presented experiments are conclusive and well controlled. I think the presentation and discussion of results are better in revised version.

The study's strength lies in the direct comparison of ClpL biochemical properties with autonomous ClpG disaggregase present in selected Gram-negative bacteria and well-studied *E. coli* system consisting of ClpB disaggregase and DnaK and its cochaperones. This puts the results in a broader context.

---

## [Referee Report · Reviewer #3 (Public Review)]

Summary:

This manuscript details the characterization of ClpL from L. monocytogenes as a potent and autonomous AAA+ disaggregase. The authors demonstrate that ClpL has potent and DnaK-independent disaggregase activity towards a variety of aggregated model substrates, and that this disaggregase activity appears to be greater than that observed with the canonical DnaK/ClpB co-chaperone. Furthermore, LmClpL appears to have greater thermostability as compared to LmDnaK, suggesting that ClpL-expressing cells may be able to withstand more severe heat stress conditions. Interestingly, LmClpL can provide thermotolerance to *E. coli* that have been genetically depleted of either ClpB or in cells expressing a mutant DnaK103. The authors further characterized the mechanisms by which ClpL interacts with protein aggregates, identifying that the N-terminal domain of ClpL is essential for disaggregase function. Lastly, by EM and mutagenesis analysis the authors report that ClpL can exist in a variety of larger macromolecular complexes, including dimer or trimers of hexamers/heptamers, and they provide evidence that the N-terminal domains of ClpL prevent dimer ring formation, thus promoting an active and substrate-binding ClpL complex. Throughout this manuscript the authors compare LmClpL to ClpG, another potent and autonomous disaggregase found in gram-negative bacteria that has been reported on previously, demonstrating that these two enzymes share homologous activity and qualities. Taken together this report clearly establishes ClpL as a novel and autonomous disaggregase.

Analysis:

The work presented in this report amounts to a significant body of novel and significant work that will be of interest to protein chaperone community. Furthermore, by providing examples of how ClpL can provide in vivo thermotolerance to both *E. coli* and *L. gasseri* the authors have expanded the significance of this work and provides novel insight into potential mechanisms responsible for thermotolerance in food-borne pathogens. The figures are clearly depicted, well-labeled, and easy to understand, and the manuscript is well-written. Experimentally the work was performed to a high standard with excellent controls, aiding in the ability for the audience to understand the major findings and conclusions. Additionally, the authors have effectively and efficiently expanded on their work through the peer review process, further increasing the understandability and significance of their work. Overall, the data presented, and analysis thereof, support the authors' conclusions, and thus this study represents an important addition to our understanding of molecular chaperone biochemistry. Lastly, this study establishes new avenues for research into autonomous disaggregates, their role in in vivo thermotolerance, and the mechanisms by which AAA+ chaperones recognize and interact with substrate proteins.

---

## [Author Response]

The following is the authors’ response to the original reviews.

**Public Reviews:**

**Reviewer #1 (Public Review):**
Summary:This work describes the mechanism of protein disaggregation by the ClpL AAA+ protein ofListeria monocytogenes. Using several model subtrate proteins the authors first show thatClpL possesses a robust disaggregase activity that does not further require the endogenous DnaK chaperone in vitro. In addition, they found that ClpL is more thermostable than the endogenous L. monocytogenes DnaK and has the capacity to unfold tightly folded protein domains. The mechanistic basis for the robust disaggregase activity of ClpL was also dissected in vitro and in some cases, supported by in vivo data performed in chaperonedeficient *E. coli* strains. The data presented show that the two AAA domains, the pore-2 site and the N-terminal domain (NTD) of ClpL are critical for its disaggregase activity. Remarkably, grafting the NTD of ClpL to ClpB converted ClpB into an autonomous disaggregase, highlighting the importance of such a domain in the DnaK-independent disaggregation of proteins. The role of the ClpL NTD domain was further dissected, identifying key residues and positions necessary for aggregate recognition and disaggregation. Finally, using sets of SEC and negative staining EM experiments combined with conditional covalent linkages and disaggregation assays the authors found that ClpL shows significant structural plasticity, forming dynamic hexameric and heptameric active single rings that can further form higher assembly states via their middle domains.Strengths:The manuscript is well-written and the experimental work is well executed. It contains a robust and complete set of in vitro data that push further our knowledge of such important disaggregases. It shows the importance of the atypical ClpL N-terminal domain in the disaggregation process as well as the structural malleability of such AAA+ proteins. More generally, this work expands our knowledge of heat resistance in bacterial pathogens.Weaknesses:There is no specific weakness in this work, although it would have helped to have a drawing model showing how ClpL performs protein disaggregation based on their new findings. The function of the higher assembly states of ClpL remains unresolved and will need further extensive research. Similarly, it will be interesting in the future to see whether the sole function of the plasmid-encoded ClpL is to cope with general protein aggregates under heat stress.

We thank the reviewer for the positive evaluation. We agree with the reviewer that it will be important to test whether ClpL can bind to and process non-aggregated protein substrates. Our preliminary analysis suggests that the disaggregation activity of ClpL is most relevant in vivo, pointing to protein aggregates as main target.

We also agree that the role of dimers or tetramers of ClpL rings needs to be further explored. Our initial analysis suggests a function of ring dimers as a resting state. It will now be important to study the dynamics of ClpL assembly formation and test whether substrate presence shifts ClpL assemblies towards an active, single ring state.

**Reviewer #2 (Public Review):**
The manuscript by Bohl et al. is an interesting and carefully done study on the biochemical properties and mode of action of potent autonomous AAA+ disaggregase ClpL from Listeria monocytogenes. ClpL is encoded on plasmids. It shows high thermal stability and provides Listeria monocytogenes food-pathogen substantial increase in resistance to heat. The authors show that ClpL interacts with aggregated proteins through the aromatic residues present in its N-terminal domain and subsequently unfolds proteins from aggregates translocating polypeptide chains through the central pore in its oligomeric ring structure. The structure of ClpL oligomers was also investigated in the manuscript. The results suggest that mono-ring structure and not dimer or trimer of rings, observed in addition to mono-ring structures under EM, is an active species of disaggregase.Presented experiments are conclusive and well-controlled. Several mutants were created to analyze the importance of a particular ClpL domain.The study's strength lies in the direct comparison of ClpL biochemical properties with autonomous ClpG disaggregase present in selected Gram-negative bacteria and well-studied *E. coli* system consisting of ClpB disaggregase and DnaK and its cochaperones. This puts the obtained results in a broader context.

We thank the reviewer for the detailed comments. There are no specific weaknesses indicated in the public review.

**Reviewer #3 (Public Review):**
Summary:This manuscript details the characterization of ClpL from L. monocytogenes as a potent and autonomous AAA+ disaggregase. The authors demonstrate that ClpL has potent and DnaKindependent disaggregase activity towards a variety of aggregated model substrates and that this disaggregase activity appears to be greater than that observed with the canonical DnaK/ClpB co-chaperone. Furthermore, Lm ClpL appears to have greater thermostability as compared to Lm DnaK, suggesting that ClpL-expressing cells may be able to withstand more severe heat stress conditions. Interestingly, Lm ClpP can provide thermotolerance to *E. coli* that have been genetically depleted of either ClpB or in cells expressing a mutant DnaK103. The authors further characterized the mechanisms by which ClpL interacts with protein aggregates, identifying that the N-terminal domain of ClpL is essential for disaggregase function. Lastly, by EM and mutagenesis analysis, the authors report that ClpL can exist in a variety of larger macromolecular complexes, including dimer or trimers of hexamers/heptamers, and they provide evidence that the N-terminal domains of ClpL prevent dimer ring formation, thus promoting an active and substrate-binding ClpL complex. Throughout this manuscript the authors compare Lm ClpL to ClpG, another potent and autonomous disaggregase found in gram-negative bacteria that have been reported on previously, demonstrating that these two enzymes share homologous activity and qualities. Taken together this report clearly establishes ClpL as a novel and autonomous disaggregase.Strengths:The work presented in this report amounts to a significant body of novel and significant work that will be of interest to the protein chaperone community. Furthermore, by providing examples of how ClpL can provide in vivo thermotolerance to both *E. coli* and *L. gasseri* the authors have expanded the significance of this work and provided novel insight into potential mechanisms responsible for thermotolerance in food-borne pathogens.Weaknesses:The figures are clearly depicted and easy to understand, though some of the axis labeling is a bit misleading or confusing and may warrant revision. While I do feel that the results and discussion as presented support the authors' hypothesis and overall goal of demonstrating ClpL as a novel disaggregase, interpretation of the data is hindered as no statistical tests are provided throughout the manuscript. Because of this only qualitative analysis can be made, and as such many of the concluding statements involving pairwise comparisons need to be revisited or quantitative data with stats needs to be provided. The addition of statistical analysis is critical and should not be difficult, nor do I anticipate that it will change the conclusions of this report.

We thank the reviewer for the valid criticism. We addressed the major concern of the reviewer and added the requested statistical analysis to all relevant figures. The analysis confirms our conclusions. We also followed the advice of the reviewer and revised axis labeling to increase clarity.

**Reviewer #1 (Recommendations For The Authors):**
It would really help to have a model showing how ClpL performs protein disaggregation based on their findings.

We show that ClpL exerts a threading activity that is fueled by ATP hydrolysis in both AAA domains and executed by pore-located aromatic residues. The basic disaggregation mechanism of ClpL therefore does not differ from ClpB and ClpG disaggregases. Similarly, the specificity of ClpL towards protein aggregates is based on simultaneous interactions of multiple N-terminal domains with the aggregate surface. We could recently describe a similar mode of aggregate recognition for ClpG [1]. We therefore prefer not to add a model to the manuscript. We are currently in preparation of a review that includes the characterization of the novel bacterial disaggregases and will present models there as we consider a review article as more appropriate for such illustrations.

AAA2 domain of ClpL in Fig 3E should be the same color as in Fig 1A.

We used light grey instead of dark grey for the ClpL AAA2 domain in Fig 3E, to distinguish between ClpL and ClpB AAA domains. This kind of illustration allows for clearer separation of both AAA+ proteins and the fusion construct LN-ClpB*. We therefore prefer keeping the color code.

Partial suppression of the dnaK mutant could be added in the main manuscript Figure.

The main figure 3 is already very dense and we therefore prefer showing respective data as part of a supplementary figure.

It would have been interesting to know if the robust autonomous disaggregation activity of ClpL would be sufficient to rescue the growth of more severe *E. coli* chaperone mutants, like dnaK tig for example. Did the authors test this?

We tested whether expression of clpL can rescue growth of *E. coli* dnaK103 mutant cells at 40°C on LB plates. This experiment is different from the restoration of heat resistance in dnaK103 cells (Figure 3, figure supplement 2A), as continuous growth at elevated temperatures (40°C) is monitored instead of cell survival upon abrupt severe heat shock (49°C). We did not observe rescue of the temperature-sensitive growth phenotype (40°C) of dnaK103 cells upon clpL expression, though expression of clpG complemented the temperature-sensitive growth phenotype (see Author response image 1 below). This finding points to differences in chaperone activities of ClpL and ClpG. It also suggests that ClpL activity is largely restricted to heat-shock generated protein aggregates, enabling ClpL to complement the missing disaggregation function of DnaK but not other Hsp70 activities including folding and targeting of newly synthesized proteins. We believe that dissecting the molecular reasons for differences in ClpG and ClpL complementation activities should be part of an independent study and prefer showing the growth-complementation data only in the response letter.

**Author response image 1. sa4fig1:** Serial dilutions (10-1 – 10-6) of E. coli dnaK103 mutant cells expressing *E. coli* dnaK, *L. monocytogenes* clpL or P. aeruginosa clpG were spotted on LB plates including the indicatedIPTG concentrations. Plates were incubated at 30°C or 40°C for 24 h. p: empty vector control.

**Reviewer #2 (Recommendations For The Authors):**
Based on results presented in Fig. 2B the authors conclude "that stand-alone disaggregases ClpL and ClpG but not the canonical KJE/ClpB disaggregase exhibit robust threading activities that allow for unfolding of tightly folded domains" (page 5 line 209). In this experiment, the threading power of disaggregases was assessed by monitoring YFP fluorescence during the disaggregation of aggregates formed by fusion luciferase-YFP protein. In my opinion, the results of the experiment depend not only on the threading power of disaggregases but also on the substrate recognition by analyzed disaggregating systems and/or processivity of disaggregases. N-terminal domain in the case of ClpL and KJE chaperones in the case of the KJE/ClpB system are involved in recognition. This is not discussed in the manuscript and the obtained result might be misinterpreted. The authors have created the LN-ClpB* construct (N-terminal domain of ClpL fused to derepressed ClpB) (Fig. 3 E and F). In my opinion, this construct should be used as an additional control in the experiment in Fig. 2 B. It possesses the same substrate recognition domain and therefore the direct comparison of disaggregases threading power might be possible.

We performed the requested experiment (new Figure 3 - figure supplement 2D). We did not observe unfolding of YFP by LN-ClpB*. Sínce ClpL and LN-ClpB* do not differ in their aggregate targeting mechanisms, this finding underlines the differences in threading power between ClpL and activated (derepressed) ClpB. It also suggests that the AAA threading motors and the aggregate-targeting NTD largely function independently.

Presented results suggest that tetramer and dimer of rings might be a "storage form" of disaggregase. It would be interesting to analyze the thermotolerance and/or phenotype of ClpL mutants that do not form tetramer and dimer (E352A). This variant possesses similar to WT disaggregation activity but does not form dimers and tetramers. If in vivo the differences are observed (for example toxicity of the mutant), the "storage form" hypothesis will be probable.

When testing expression of clpL-MD mutants (E352A, F354A), which cannot form dimers and tetramers of ClpL rings, in *E. coli* ∆clpB cells, we observed reduced production levels as compared to ClpL wildtype and speculated that reduced expression might be linked to cellular toxicity. We therefore compared spotting efficiencies of *E. coli* ∆clpB cells expression clpL, ∆NclpL or the clpL-MD mutants at different temperatures. Expression of clpL at high levels abrogated colony formation at 42°C (new Figure 6 - figure supplement 3). ClpL toxicity was dependent on its NTD as no effect was observed upon expression of ∆N-clpL. ClpL-MD mutants (E352A, F354A) were expressed at much lower levels and exhibited strongly increased toxicity as compared to ClpL-WT when produced at comparable levels (new Figure 6 – figure supplement 3). This implies a protective role of ClpL ring dimers and tetramers in the cellular environment by downregulating ClpL activity. We envision that the formation of ClpL assemblies restricts accessibility of the ClpL NTDs and reduces substrate interaction. Increased toxicity of ClpL-E352A and ClpL-F354A points to a physiological relevance of the dimers and tetramers of ClpL rings and is in agreement with the proposed function as storage forms. We added this potential role of ClpL ring assemblies to the discussion section. Due to the strongly reduced production levels of ClpL MD mutants and their enhanced toxicity at elevated temperatures we did not test for their ability to restore thermotolerance in *E. coli* ∆clpB cells.

Figure 6G and Figure 6 -figure supplement 2 - it is not clear what is the difference in the preparation of WT and WTox forms of ClpL.

ClpL WT was purified under reduced conditions (+ 2 mM DTT), whereas WTox was purified in absence of DTT, thus serving as control for ClpL-T355C, which forms disulfide bonds upon purification without DTT. We have added respective information to the figure legend and the materials and methods section.

Page 5 line 250 - wrong figure citation. Instead of Figure 1 - Figure Supplement 2A should be Figure 3 - Figure Supplement 2A.Page 5 line 251 - wrong figure citation. Instead of Figure 1 - Figure Supplement 2B/C should be Figure 3 - Figure Supplement 2B/C.Page 7 line 315 - wrong figure citation. Instead of Figure 4F, it should be Figure 4G Figure 1 - Figure Supplement 2E - At first glance, this Figure does not correspond to the text and is confusing. It would be nice to have bars for Lm ClpL activity in the figure. Alternatively, the description of the y-axis might be changed to "relative to Lm ClpL disaggregation activity" instead of "relative disaggregation activity". One has to carefully read the figure legend to find out that 1 corresponds to Lm ClpL activity.

We have corrected all mistakes and changed the description of y-axis (Figure 1 - figure Supplement 2E) as suggested.

**Reviewer #3 (Recommendations For The Authors):**
(1) While the authors make many experimental comparisons throughout their study, no statistical tests are described or presented with their results or figures, nor are these statistical tests described in the methods. While the data as presented does appear to support the author's conclusions, without these statistical tests no meaningful conclusions from paired analysis can be drawn. Critically, please report these statistical tests. As a general suggestion please include the statistics (p-values) in the results section when presenting this data, as well as in the figure legends, as this will allow the reader to better understand the authors' presentation and interpretation of the data.

We have added statistical tests to all relevant figures. The analysis is confirming our former statements. We have further clarified our approach for the statistical analysis in the methods section. We report p-values in the results section, however, due to the volume of comparisons we did not add individual p-values to the figure legends but used standard labeling with stars.

(2) Some of the axis labels for the presented graphs are a bit misleading or confusing. Many describe a relative (%) disaggregation rate, but it is not clear from the methods or figure legends what this rate is relative to. Is it relative to non-denatured substrates, to no chaperone conditions, etc.? Is it possible to present the figures with the raw data rates/activity (ex. luciferase activity / time) vs. relative rates? I think that labeling these figure axes with "disaggregation rate" is a bit misleading as none of these experiments measure the actual rate of disaggregation of these model substrates per se (say by SEC-MALS or other biophysical measurements), but instead infer the extent of disaggregation by measuring a property of these substrates, i.e. luciferase activity or fluorescence intensity over time. Thus, labeling these figures with the appropriate axis for what is being measured, and then clarifying in the methods and results what is being inferred by these measurements, will help solidify the author's conclusions.

Relative (%) disaggregation rate usually refers to the disaggregation activity of ClpL wildtype serving as reference. We clarified this point in the revised text and respective figure legends. We now also refer to the process measured (e.g. relative refolding activity of aggregated Luciferase instead of relative disaggregation activity) as suggested by the reviewer and added clarifications to text and materials and methods.

Since we have many measurements for our most frequently used assays and have a reasonable estimate for the general variance within these assays, we found it reasonable to show activity data in relation to fixed controls. This reduces the impact of unspecific variance and thereby makes more accurate comparisons between different repetitions. The reference is now indicated in the axis title.

(3) The figures are well presented, clutter-free, and graphically easy to understand. Figure legends have sufficient information aside from the aforementioned statistical information and should include the exact number of independent replicates for each panel/experiment (ex. n=4), not just a greater than 3. While the figures do show each data point along with the mean and error, in some figures it is difficult to determine the number of replicate data points. Example figures 2c, 2d, and 3a. Also, please state whether the error is std. error or SEM.

While we agree, that this is valuable information, we fear that overloading the figure legends with information may take a toll on the readability. We therefore decided to append the number of replicates for each experiment in a separate supplementary table (Table S2). The depicted error is showing the SD and not the SEM, which we also specified in the figure legends.

(4) There are various examples throughout the results where qualitative descriptors are used to describe comparisons. Examples of this are "hardly enhanced" (Figure 1) and "partially reduced" (Figure 6). While this is not necessarily wrong, qualitative descriptions of comparisons in this manner would require further explanation. What is the definition of "hardly" or "partially"? My recommendation is to just state the data quantitatively, such as "% enhanced" or "reduced by x", this way there is no misinterpretation. Examples of this can be found in Figures 6C-G. This would require a full statistical overview and presentation of these stats in the results.

We followed the reviewer`s advice and no longer use the terms criticized (e.g. “hardly enhanced”). We instead provide the requested quantifications in the text.

Questions for Figures:

Figures 1B and 1C:

(1) Is the disaggregase activity of ClpL towards heat-denatured luciferase and GFP ATPdependent? While the authors later in the manuscript show that mutations within the Walker B domains dramatically impair reactivation (disaggregation) of denatured luciferase, this does not rule out an ATP-independent effect of these mutations. Thus, the authors should test whether disaggregase activity is observed when wild-type ClpL is incubated with denatured substrates without ATP present or in the presence of ADP only.

We tested for ClpL disaggregation activity in absence of nucleotide and presence of ADP only(new Figure 1 – figure supplement 2A). We did not observe any activity, demonstrating that ClpL activity depends on ATP binding and hydrolysis (see also Figure 3 – figure supplement 1D: ATPase-deficient ClpL-E197A/E530A is lacking disaggregation activity).

(2) The authors suggest that a reduction in disaggregase activity observed in samples combining Lm ClpL and KJE (Figure 1C, supp. 1C-E) could be due to competition for protein aggregate binding as observed previously with ClpG. Did the authors test this directly by pulldown assay or another interaction-based assay? While ClpL and ClpG appear to work in a similar manner, it would be good to confirm this. Also, clarification on how this competition operates would be useful. Is it that ClpL prevents aggregates from interacting with KJE, or vice versa?

We probed for binding of ClpL to aggregated Malate Dehydrogenase in the presence of L. monocytogenes or *E. coli* Hsp70 (DnaK + respective J-domain protein DnaJ) by a centrifugation-based assay. Here, we used the ATPase-deficient ClpL-E197A/E530A (ClpLDWB) mutant, ensuring stable substrate interaction in presence of ATP. We observe reduced binding of ClpL-DWB to protein aggregates in presence of DnaK/DnaJ (new Figure 1 – figure supplement 2G). This finding indicates that both chaperones compete for binding to aggregated proteins and explains inhibition of ClpL disaggregation activity in presence of Hsp70.

(3) Related to the above, while incubation of aggregated substrates with ClpL and KJE does appear to reduce aggregase activity towards GFP (Figure 1c), α-glucosidase (Supp. 1C), and MDH (Supp. 1D), this doesn't appear to be the case towards luciferase (Figure 1b, Supp. 1b). Furthermore, ClpL aggregase activity is reduced towards luciferase when combined with *E. coli* KJE (Supp. 1e) but not with Lm KJE (Figure 1b). The authors provide no commentary or explanation for these observations. Furthermore, these results complicate the concluding statement that "combining ClpL with Lm KJE always led to a strong reduction in disaggregation activity ... ".

We suggest that the differing inhibitory degrees of the KJE system on ClpL disaggregation activities reflect diverse binding affinities of KJE and ClpL to the respective aggregates. While we usually observe strong inhibition of ClpL activity in presence of KJE, this is different for aggregated Luciferase. This points to specific structural features of Luciferase aggregates or the presence of distinct binding sites on the aggregate surface that favour ClpL binding. We have added a respective comment to the revised manuscript.

The former statement that “combining ClpL with Lm KJE always led to a strong reduction in disaggregation activity” referred to aggregated GFP, MDH and α-Glucosidase for which a strong inhibition of ClpL activity was observed. We have specified this point.

Figures 1D and 1E:(1) The authors conclude that the heat sensitivity of ΔClpL L. gasseri cells is because they do not express the canonical ClpB disaggregase. A good test to validate this would be to express KJE/ClpB in these Lg ΔClpL cells to see if heat-sensitivity could be fully or partially rescued.

We agree that such experiment would further strengthen the in vivo function of ClpL as alternative disaggregase. However, such approach would demand for co-expression of *E. coli* ClpB with the authentic *E. coli* DnaK chaperone system (KJE), as ClpB and DnaK cooperate in a species-specific manner [2-4]. This makes the experiment challenging, also because the individual components need to be expressed at a correct stochiometry. Furthermore, the presence of the authentic L. gasseri KJE system, which is likely competing with the *E. coli* KJE system for aggregate binding, will hamper *E. coli* KJE/ClpB disaggregation activity in L. gasseri. In view of these limitations, we would like to refrain from conducting such an experiment.

(2) The rationale for investigating Lg ClpL, and the aggregase activity assays are compelling and support the hypothesis that ClpL contributes to thermotolerance in multiple grampositive species. Though, from Figure 1d, why was only Lg ClpL investigated? It appears that S. thermophilus also lacks the canonical ClpB disaggregase and demonstrates ΔClpL heat sensitivity. There is also other Lactobacillus sp. presented that lack ClpB but were not tested for heat sensitivity. Why only test and move forward with L. gasseri? Lastly, L. mesenteroides is ClpB-negative but doesn't demonstrate ΔClpL heat sensitivity. Why?

We wanted to document high, partner-independent disaggregation activity for another ClpL homolog. We chose L. gasseri, as (i) this bacterial species lacks a ClpB homolog and (ii) a ∆clpL mutant exhibit reduced survival upon severe heat shock (thermotolerance phenotype), which is associated with defects in cellular protein disaggregation. The characterization of L. gasseri ClpL as potent disaggregase in vitro represents a proof-of-concept and allows to generalize our conclusion. We therefore did not further test S. thermophilus ClpL. L. mesenteroides encodes for ClpL but not ClpB, yet, a ∆clpL mutant has not yet been characterized in this species to the best of our knowledge. As we wanted to link ClpL in vitro activity with an in vivo phenotype, we did not characterize L. mesenteroides ClpL.

We agree with the reviewer that the characterization of additional ClpL homologs is meaningful and interesting, however, we strongly believe that such analysis should be part of an exhaustive and independent study.

Figures 2A and 2B:(1) Figure 2B demonstrates that both ClpL and ClpG, but not the canonical KJE/ClpB, are able to unfold YFP during the luciferase disaggregation process, suggesting that ClpL and ClpG exhibit stronger threading activity. A technical question, can luciferase activity be measured alongside in the same assay sample? If so, would you expect to observe a concomitant increase in luciferase activity as YFP fluorescence decreases?

KJE/ClpB can partially disaggregate and refold aggregated Luciferase-YFP without unfolding YFP during the disaggregation reaction [5]. YFP unfolding is therefore not linked to refolding of aggregated Luciferase-YFP. On the other hand, unfolding of YFP during disaggregation can hamper the refolding of the fused Luciferase moiety as observed for the AAA+ protein ClpC in presence of its partner MecA [5]. These diverse effects make the interpretation of LuciferaseYFP refolding experiments difficult as the degree of YFP unfolding activity does not necessarily correlate with the extend of Luciferase refolding. We therefore avoided to perform the suggested experiment.

Figure 2C and 2D:(1) Thermal shift assays for ClpL, ClpG, and DnaK were completed with various nucleotides. Were these experiments also completed with samples in their nucleotide-free apo state? Also, while all these chaperones are ATPases, the nucleotides used differ, but no explanation is provided. Comparison should be made of these ATPases bound to the same molecules.

We did not monitor thermal stabilities of chaperones without nucleotide as such state is likely not relevant in vivo. We used ATPγS in case of ClpL to keep the AAA+ protein in the ATPconformation. ATP would be rapidly converted to ADP due to the high intrinsic ATPase activity of ClpL. In case of DnaK ATPγS cannot be used as it does not induce the ATP conformation [6]. The low intrinsic ATPase activity of DnaK allows determining the thermal stability of its ATP conformation in presence of ATP. This is confirmed by calculating a reduced thermal stability of ADP-bound DnaK.

(2) The authors suggest that incubation at 55⁰C will cause unfolding of Lm DnaK, but not ClpL, providing ClpL-positive Lm cells disaggregase activity at 55⁰C. While the thermal shift assays in Figures 2C and 2D support this, an experiment to test this would be to heat-treat Lm DnaK and ClpL at 55⁰C then test for disaggregase activity using either aggregated luciferase or GFP as in Figure 1.

We followed the suggestion of the reviewer and incubated Lm ClpL and DnaK at 55-58°C in presence of ATP for 15 min prior to their use in disaggregation assays. We compared the activities of pre-heated chaperones with controls that were incubated at 30°C for 15 min. Notably, we did not observe a loss of DnaK disaggregation activity, suggesting that thermal unfolding of DnaK at this temperature is reversible. We provide these data as Figure 2 -figure supplement 1 and added a respective statement to the revised manuscript.

Figure 3B:(1) The authors state that ATPase activity of ΔN-ClpL was "hardly affected", but from the data provided it appeared to result in an approximate 35% reduction. As discussed above, no stats are provided for this figure, but given the error bars, it is highly likely that this reduction is significant. Please perform this statistical test, and if significant, please reflect this in the written results as well as the figure. Lastly, if this reduction in ATPase activity is significant, why would this be so, and could this contribute to the reduction in aggregase activity towards luciferase and MDH observed in Figure 3A?

We applied statistical tests as suggested by the reviewer, showing that the reduction in ATPase activity of ∆N-ClpL is statistically significant. N-terminal domains of Hsp100 proteins can modulate ATPase activity as shown for the family member ClpB, functioning as auxiliary regulatory element for fine tuning of ClpB activity [7]. We speculate that the impact of the ClpL-NTD on the assembly state (stabilization of ClpL ring dimers) might affect ClpL ATPase activity. We would like to point out that other ClpL mutants (e.g. NTD mutant ClpL-Y51A; MDmutant ClpL-F354A) have a similarly reduced ATPase activity, yet exhibit substantial disaggregation activity (approx. 2-fold reduced compared to ClpL wildtype). In contrast ∆NClpL does not exhibit any disaggregation activity. This suggests that the loss of disaggregation activity is caused by a substrate binding defect but not by a partial reduction in ATPase activity. We added a comment on the reduced ATPase activity and also discuss its potential reasons in the discussion section.

(2) I think the authors' conclusion that deletion of the ClpL NTD does not contribute to structural defects of ClpL is premature given the apparent reduction in ATPase activity. Did the authors perform any biophysical analysis of ΔN-ClpL to confirm this conclusion? Thermal shift assays, Native-PAGE, or size-exclusion chromatography for aggregates would all be good assays to demonstrate that the wild-type and ΔN-ClpL have similar structural properties. Surprisingly, Figure 6 describes significant macromolecular changes associated with ΔN-ClpL such that it preferentially forms a dimer of rings. Furthermore, in Supp. Figure 6D the authors report that ΔN-ClpL appears to have an increased Tm as compared to WT- or ΔM-ClpL. The authors should reflect these observations as deletion of the ClpL NTD does appear to contribute to structural changes, though perhaps only at the macromolecular scale, i.e. dimerization of the rings.

We have characterized the oligomeric state of ∆N-ClpL by size exclusion chromatography (Figure 6 – figure supplement 1A) and negative staining electron microscopy (Figure 6C), both showing that it forms assemblies similar to ClpL wildtype. We did not observe an increased tendency of ∆N-ClpL to form aggregates and the protein remained fully soluble after several cycles of thawing and freezing. EM data reveal that ∆N-ClpL exclusively form ring dimers, suggesting that the NTDs destabilize MD-MD interactions. The stabilized interaction between two ∆N-ClpL rings can explain the increased thermal stability (Figure 6 – figure supplement 1D). We speculate that the ClpL NTDs either affect MD-MD interactions through steric hindrance or by directly contacting MDs. We have added a respective statement to the discussion section.

Figure 3C and 3D:(1) Given the larger error in samples expressing ClpG (100) or ClpL (100) statistical analysis with p-values is required to make conclusions regarding the comparison of these samples vs. plasmid-only control. The effect of ΔN-ClpL vs. wild-type ClpL looks compelling and does appear to attenuate the ClpL-induced thermotolerance. This is nicely demonstrated in Figure 3D.

We quantified respective spot tests (new Figure 3E) and tested for statistical significance as suggested by the reviewer. We show that restoration of heat resistance is significant for the first 30 min. While we always observe rescue at later timepoints significance is lost here due to larger deviations in the number of viable cells and thus the degree of complementation.

Figure 3F:(1) What is the role of the ClpB NTD? It appears to be dispensable for disaggregase activity, assuming that ClpB is co-incubated with KJE. A quick explanation of this domain in ClpB could be useful.

The ClpB NTD is not required for disaggregation activity, as ClpB is recruited to protein aggregates by DnaK, which interacts with the ClpB MDs. Still, two functions have been described for the ClpB NTD. First, it can bind soluble unfolded substrates such as casein [8]. This substrate binding function can increase ClpB disaggregation activity towards some aggregated model substrates (e.g. Glucose-6-phosphate dehydrogenase) [9]. However, NTD deletion usually does not decrease ClpB disaggregation activity and can even lead to an increase [7, 10, 11]. An increased disaggregation activity of ∆N-ClpB correlates with an enhanced ATPase activity, which is explained by NTDs stabilizing a repressing conformation of the ClpB MDs, which function as main regulators of ClpB ATPase activity [7]. We added a short description on the role of the ClpB NTD to the respective results section.

(2) The result of fusing the ClpL NTD to ClpB supports a role for this NTD in promoting autonomous disaggregase activity. What would you expect to observe if the fused Ln-ClpB protein was co-incubated with KJE? Would this further promote disaggregase activity, or potentially impair through competition? This experiment could potentially support the authors' hypothesis that ClpL and ClpB/KJE can compete with each other for aggregated substrates as suggested in Figure 1.

We have performed the suggested experiment using aggregated MDH as model substrate. We did not observe an inhibition of LN-ClpB* disaggregation activity in presence of KJE. In contrast ClpL disaggregation activity towards aggregated MDH is inhibited upon addition of KJE due to competition for aggregate binding (Figure 1 – figure supplement 2D/F). Disaggregation activity of LN-ClpB* in presence of KJE can be explained by functional cooperation between both chaperone systems, which involves interactions between aggregate-bound DnaK and the ClpB MDs of the LN-ClpB* fusion construct. We prefer showing these data only in the response letter but not including them in the manuscript, as respective results distract from the main message of the LN-ClpB* fusion construct: the ClpL NTD functions as autonomous aggregatetargeting unit that can be transferred to other Hsp100 family members.

**Author response image 2. sa4fig2:** LN-ClpB* cooperates with DnaK in protein disaggregation. Relative MDH disaggregation activities of indicated disaggregation systems were determined. KJE: DnaK/DnaJ/GrpE. The disaggregation activity of Lm ClpL was set to 1. Statistical Analysis: Oneway ANOVA, Welch’s Test for post-hoc multiple comparisons. Significance levels: ***p < 0.001. n.s.: not significant.

Figures 4E and 4F:(1) While the effect of various NTD mutations follows a similar trend in regard to the impairment of ClpL-mediated disaggregation of luciferase and MDH, the degree of these effects does appear different. For example, patch A and C mutations reduce ClpL disaggregase activity towards luciferase (~60% / 50% reduction) vs. MDH (>90%) respectively. While these results do suggest a critical role for residues in patches A and C of ClpL, these substrate-specific differences are not discussed. Why would we expect a difference in the effect of these patch A/C ClpL mutations on different substrates?

We speculate that the aggregate structure and the presence or distributions of ClpL NTD binding sites differ between aggregated Luciferase and MDH. A difference between both aggregated model substrates was also observed when testing for an inhibitory effect of Lm KJE (and Ec KJE) on ClpL disaggregation activity (see comment above). We speculate that the mutated NTD residues make specific contributions to aggregate recognition. The severity of binding defects (and reduction of disaggregation activities) of these mutants will depend on specific features of the aggregated model substrates. We now point out that ClpL NTD patch mutants can differ in disaggregation activities depending on the aggregated model substrate used and refer to potential differences in aggregate structures.

(2) The authors suggest that the loss of disaggregation activity of selected NTD mutants could be linked to reduced binding to aggregated luciferase. While this is likely given that these mutations do not appear to affect ATPase activity (Supp. 4), it could be possible that these mutants can still bind to aggregated luciferase and some other mechanism may impair disaggregation. A pull-down assay would help to prove whether reduced binding is observed in these NTD ClpL mutants. This also needs to be confirmed for Supp. Figure 4.2H.

We have shown a strong correlation between loss of aggregate binding and disaggregation activity for several NTD mutants (Fig. 4G, Figure 4 – figure supplement 2H). We decided to perform the aggregate binding assay only with mutants that show a full but not a partial disaggregation defect as we made the experience that the centrifugation-based assay provides clear and reproducible results for loss-of-activity mutants but has limitations in revealing differences for partially affected mutants. This might be explained by the use of nonhydrolyzable ATPγS in these experiments, which strongly stabilizes substrate interactions, potentially covering partial binding defects. We agree with the reviewer that some ClpL NTD mutants might have additional effects on disaggregation activity by e.g. controlling substrate transfer to the processing pore site. We have added a respective comment to the revised manuscript.

(3) Supp. Figure 4.2H has no description in the figure legend. The Y-axes states % aggregate bound to chaperone. How was this measured? See the above comments for Figures 4E and4F.

We apologize and added the description to the figure legend. The determination of % aggregate bound chaperone is based on the quantifications of chaperones present in the supernatant and pellet fractions after sample centrifugation. Background levels of chaperones in the pellet fractions in absence of protein aggregates were subtracted. We added this information to the materials and methods section.

Figure 6G:The authors observed reduced disaggregase activity and ATPase activity of mutant T355C under both oxidative and reducing conditions. While this observation under oxidative conditions supports the authors' hypothesis, under reducing conditions (+DTT) we would expect the enzyme to behave similarly to wild-type ClpL unless this mutation has other effects. Can the authors please comment on this and provide an explanation or hypothesis?

The reviewer is correct, ClpL-T355C exhibit a reduced disaggregation activity (Figure 6 – figure supplement 2B). We observe a similar reduction in disaggregation activity for the ClpL MD mutant F354A, pointing to an auxiliary function of the MD in protein disaggregation. We have made a respective comment in the discussion section of the revised manuscript. How exactly ClpL MDs support protein disaggregation is currently unclear and will be subject of future analysis in the lab. We strongly believe that such analysis should be part of an independent study.

Discussion:In the fourth feature, it is discussed that one disaggregase feature of ClpL is that it does not cooperate with the ClpP protease. While a reference is provided for the canonical ClpB, no data in this paper, nor a reference, is provided demonstrating that ClpL does not interact with ClpP. As discussed, it is highly unlikely that ClpL interacts with ClpP given that ClpL does not contain the IGL/F loops that mediate the interaction of ClpP with cochaperones, such as ClpX, but data or a reference is needed to make such a factual statement.

The absence of the IGL/F loop makes an interaction between ClpL and ClpP highly unlikely. However, the reviewer is correct, direct evidence for a ClpP-independent function of ClpL, though very likely, is not provided. We have therefore rephrased the respective statement: “Forth, novel disaggregases lack the specific IGL/F signature motif, which is essential for cooperation of other Hsp100 proteins with the peptidase ClpP. This feature is shared with the canonical ClpB disaggregase [12] suggesting that protein disaggregation is primarily linked to protein refolding.”.

References

(1) Katikaridis P, Simon B, Jenne T, Moon S, Lee C, Hennig J, et al. Structural basis of aggregate binding by the AAA+ disaggregase ClpG. J Biol Chem. 2023:105336.

(2) Glover JR, Lindquist S. Hsp104, Hsp70, and Hsp40: A novel chaperone system that rescues previously aggregated proteins. Cell. 1998;94:73-82.

(3) Krzewska J, Langer T, Liberek K. Mitochondrial Hsp78, a member of the Clp/Hsp100 family in *Saccharomyces cerevisiae*, cooperates with Hsp70 in protein refolding. FEBS Lett. 2001;489:92-6.

(4) Seyffer F, Kummer E, Oguchi Y, Winkler J, Kumar M, Zahn R, et al. Hsp70 proteins bind Hsp100 regulatory M domains to activate AAA+ disaggregase at aggregate surfaces. Nat Struct Mol Biol. 2012;19:1347-55.

(5) Haslberger T, Zdanowicz A, Brand I, Kirstein J, Turgay K, Mogk A, et al. Protein disaggregation by the AAA+ chaperone ClpB involves partial threading of looped polypeptide segments. Nat Struct Mol Biol. 2008;15:641-50.

(6) Theyssen H, Schuster H-P, Bukau B, Reinstein J. The second step of ATP binding to DnaK induces peptide release. J Mol Biol. 1996;263:657-70.

(7) Iljina M, Mazal H, Goloubinoff P, Riven I, Haran G. Entropic Inhibition: How the Activity of a AAA+ Machine Is Modulated by Its Substrate-Binding Domain. ACS chemical biology. 2021;16:775-85.

(8) Rosenzweig R, Farber P, Velyvis A, Rennella E, Latham MP, Kay LE. ClpB N-terminal domain plays a regulatory role in protein disaggregation. Proc Natl Acad Sci U S A. 2015;112:E6872-81.

(9) Barnett ME, Nagy M, Kedzierska S, Zolkiewski M. The amino-terminal domain of ClpB supports binding to strongly aggregated proteins. J Biol Chem. 2005;280:34940-5.

(10) Beinker P, Schlee S, Groemping Y, Seidel R, Reinstein J. The N Terminus of ClpB from Thermus thermophilus Is Not Essential for the Chaperone Activity. J Biol Chem. 2002;277:47160-6.

(11) Mogk A, Schlieker C, Strub C, Rist W, Weibezahn J, Bukau B. Roles of individual domains and conserved motifs of the AAA+ chaperone ClpB in oligomerization, ATP-hydrolysis and chaperone activity. J Biol Chem. 2003;278:15-24.

(11) Weibezahn J, Tessarz P, Schlieker C, Zahn R, Maglica Z, Lee S, et al. Thermotolerance Requires Refolding of Aggregated Proteins by Substrate Translocation through the Central Pore of ClpB. Cell. 2004;119:653-65.